**Brief Communication**

# Multiplexed transcriptome discovery of RNA-binding protein binding sites by antibody-barcode eCLIP

Daniel A. Lorenz [1], Hsuan-Lin Her[2,3,4], Kylie A. Shen[1], Katie Rothamel[2,3,4], Kasey R. Hutt[1], Allan C. Nojadera [1], Stephanie C. Bruns[1], Sergei A. Manakov[1], Brian A. Yee[2,3,4], Karen B. Chapman [1] ✉ & Gene W. Yeo [2,3,4] ✉

Ultraviolet crosslinking and immunoprecipitation (CLIP) methodologies enable the identification of RNA binding sites of RNA-binding proteins (RBPs). Despite improvements in the library preparation of RNA fragments, the enhanced CLIP (eCLIP) protocol requires 4 days of hands-on time and lacks the ability to process several RBPs in parallel. We present a new method termed antibody-barcode eCLIP that utilizes DNA-barcoded antibodies and proximity ligation of the DNA oligonucleotides to RBP-protected RNA fragments to interrogate several RBPs simultaneously. We observe performance comparable with that of eCLIP with the advantage of dramatically increased scaling while maintaining the same material requirement of a single eCLIP experiment.

RBPs are critical regulators of gene expression, controlling the rate, location and timing of RNA maturation[1–4]. As such, dysregulation of RBP function is associated with diverse genetic and somatic disorders, such as neurodegeneration and cancer[5,6]. To uncover the molecular mechanisms by which RBPs affect RNA processing, technologies such as RNA immunoprecipitation (RIP) and CLIP facilitate transcriptome-wide RNA binding site identification[7–10]. eCLIP enabled the generation of 223 eCLIP datasets profiling targets for 150 RBPs in K562 and HepG2 cell lines via a standardized protocol[11]. These target maps revealed unexpected principles of RNA processing[11–13]. However, the number of protein-coding genes with experimental or computational evidence for RNA-binding properties continues to increase, accounting for at least around 15% of the human genome[14–17], and our ENCODE pilot still represents less than 10% of annotated RBPs.

We opine that reducing the technical complexity of the eCLIP protocol is pivotal to accelerating our progress toward an exhaustive characterization of RBPs. Two main limitations to scaling remain. First, all current CLIP-based methods feature SDS–PAGE and a nitrocellulose membrane transfer step to size-select for the immunoprecipitated protein–RNA complex[18–20]. This manual excision of estimated protein–RNA

bands is tedious, requires an additional 1.5–2 days and is vulnerable to user-to-user variation. Second, each individual RBP requires a separate immunoprecipitation (IP) step, which places a burden on the quantity of input material required for studying many RBPs.

Here, we develop antibody-barcode eCLIP (ABC). Our optimizations address both of the constraints of eCLIP through the incorporation of DNA-barcoded antibodies that allow on-bead proximity-based ligations to replace the SDS–PAGE and membrane transfer steps. The DNA barcodes also enable the identity of different RBPs within the same sample to be distinguished, dramatically reducing the input requirement per RBP (Fig. 1a and Extended Data Fig. 1). We evaluated ABC using two well-characterized RBPs, RNA Binding Fox-1 Homolog 2 (RBFOX2), which recognizes GCAUG motifs, and the Stem-Loop Binding Protein (SLBP), which interacts specifically with histone mRNAs. After observing no change in IP efficiency after antibody barcoding, (Supplementary Fig. 1) duplicate singleplex ABC experiments for RBFOX2 were performed in HEK293XT cells. SLBP was similarly interrogated in K562 cells. Reads were mapped and processed as previously described[11,21]. We first compared the library complexity as a surrogate measure of library efficiency by enumerating the number of 'usable'

[1]Eclipse Bioinnovations, San Diego, CA, USA. [2]Department of Cellular and Molecular Medicine, University of California San Diego, La Jolla, CA, USA. [3]Institute for Genomic Medicine, University of California San Diego, La Jolla, CA, USA. [4]Stem Cell Program, University of California San Diego, La Jolla, CA, USA. ✉e-mail: karen.chapman@eclipsebio.com; geneyeo@ucsd.edu

reads, defined as reads that map uniquely to the genome and remain after discarding PCR duplicates. We observed that eCLIP and ABC exhibited similar library complexity (Extended Data Fig. 2a,b). Library yield was dependent on UV crosslinking as a no UV control displayed a 32-fold decrease by quantitative PCR (qPCR) (Extended Data Fig. 2c). Examination of individual binding sites revealed comparable read density between ABC and eCLIP at RBFOX2 (for example, intronic region of NDEL1) and SLBP (for example, 3′ untranslated region (UTR) of H1−2) binding sites (Fig. 1b)[12].

To evaluate ABC with a transcriptome-wide view, we initially focused on RBFOX2 and observed that peaks from ABC showed similar enrichments to downsampled eCLIP data in proximal and distal introns (Extended Data Fig. 3a) and were significantly enriched for the RBFOX2 motif (Extended Data Fig. 3b). Reproducible peaks obtained from irreproducible discovery rate (IDR) analysis of RBFOX2 eCLIP data serve as empirically defined RBFOX2 sites. We observed that the proportion of ABC reads present within reproducible RBFOX2 peaks also mirrors eCLIP (Fig. 1c). We also compared the fraction of reads that contained the conserved GCAUG sequence, as evolutionarily sequence conserved RBFOX2 motifs are more likely to be authentic sites[22] (Supplementary Note 1). We observed that the fraction of reads that contain the conserved motif is similar (around 0.38% for eCLIP, and around 0.4% for ABC; Fig. 1c). As RBFOX2 exhibits positional dependencies in its regulation of alternative splicing[22], we demonstrated that ABC-derived peaks reproduced the eCLIP enrichment for RBFOX2 binding upstream and within exons that are included in the mature mRNA exons; as well as binding downstream to enhance exon recognition and exclusion from mature mRNA (Fig. 1d). Next, we shifted our focus to SLBP. Both ABC and eCLIP displayed a similar percent of reads that map to histone RNAs (Fig. 1e). Metagene analysis also revealed a peak at the well-characterized stem-loop within the 3′ UTR of histone mRNAs (Fig. 1f). To compare the gene level enrichment, we ranked genes by the most enriched peaks after normalization and identified the top 100 genes in each dataset. Both technologies exhibited similar enrichment of histone genes (Fig. 1g and Supplementary Table 1). Our comparison of ABC and eCLIP analyses for the RBPs RBFOX2 and SLBP suggests that ABC performs with comparable sensitivity and specificity to eCLIP at both read and peak-level features.

A defining advantage of ABC over current CLIP-based methodologies is that several RBPs can be interrogated simultaneously from a single sample (Fig. 2a). To demonstrate this key feature, in addition to RBFOX2, we selected nine other RBPs previously characterized by ENCODE3 (ref. [11]) in K562 cells to exhibit a diversity of known binding preferences within genic regions indicative of function: DDX3 and EIF3G in the 5′ UTR; IGF2BP2, FAM120A, PUM2 and ZC3H11A in the 3′ UTR; LIN28B in the CDS; SF3B4 in branch point recognition at the 3′ splice site and PRPF8, which is downstream of the 5′ splice site. We performed triplicate, multiplexed ABC experiments after conjugating barcoded oligonucleotides to each RBP specific antibody. These antibodies were previously validated by ENCODE[7,11]. After computational deconvolution of the barcodes, we processed each RBP within each ABC sample separately. For each RBP, we removed reads that map to repetitive elements, only retaining reads that mapped uniquely to the human genome. As each antibody in a multiplexed reaction immunoprecipitates different amounts of protein–RNA targets due to factors

such as protein expression and expression levels of RNA targets, we expectedly observe nonuniform coverage across barcodes. For a fair comparison, we computationally downsampled the uniquely mapping eCLIP reads to the same sequencing depth as the demultiplexed ABC libraries (Supplementary Table 2 and Supplementary Note 2) and observed similar library complexity (Extended Data Fig. 4).

Peak-calling by CLIPper[23] was performed with the same parameters on the ABC and eCLIP samples. The numbers of initial peaks were similar between ABC and eCLIP (Extended Data Fig. 5). To identify statistically significantly enriched peaks over background, the eCLIP protocol incorporates a size-matched input (SMI) control representing all RBP−RNA interactions (including the interrogated RBP) in the same migratory range on the gel and membrane to capture nonspecific background RNAs[12]. However, as ABC removed the gel and membrane transfer steps, we reasoned that an alternative but related measure of the specificity for an RBP on a binding site is achieved by leveraging the binding information from all (other) RBPs in the multiplexed reaction as a 'complement' control (CC; Supplementary Fig. 2). To do so, we computed the chi-squared statistic using the observed number of reads within the region specified by a given RBP peak, relative to the total number of reads for that RBP, compared with the number of reads from the other nine RBPs within the same region, relative to the total number of reads for those RBPs. ABC peaks that satisfied thresholds of $P < 0.001$ computed with the chi-square statistic and were greater than eightfold higher over CC were deemed enriched peaks for that RBP. For the four RBPs, RBFOX2, PUM2, PRPF8 and SF3B4, which have well-characterized motifs, HOMER[24] identified their respective motifs from the ABC enriched peaks de novo (Extended Data Fig. 6).

Next, we observed that the enriched peaks from ABC had similar distributions across genic regions to their eCLIP counterparts (Fig. 2b), with an equivalent number of total peaks (Extended Data Fig. 5). We also compared the differences in the representation of genic features among the enriched peaks when we utilized total RNA-seq[25,26]. Using CC to compute statistical significance and fold changes resulted in improved ranking of the genic regions known to be preferred by the RBP (Extended Data Fig. 7a). For example, intronic regions are better represented among the highly ranked peaks for the splicing factor RBFOX2, compared with using RNA-seq as background. Furthermore, compared with RNA-seq as background, we found that peaks prioritized by CC have fewer overlaps with discarded eCLIP peaks (using SMI as background) (Extended Data Fig. 7b,c). This suggests that CC accounts for the experimental background better than total RNA-seq. Interestingly, when all the RBPs in the multiplex set (which did not contain SLBP) were used as the CC in analysis of the SLBP singleplex experiment, peaks in histone RNAs are prioritized higher in rank, compared with using RNA-seq as background (Extended Data Fig. 7d,e).

To analyze the peak locations with higher resolution around splice sites, we plotted the metagene profiles of the enriched peaks for the spliceosomal proteins SF3B4 and PRPF8. Both RBPs displayed strong positional preferences proximal to their respective splice sites (Fig. 2c). We observed that the ABC-derived peaks (CC) for PRPF8 were closer to the annotated 5′ splice sites than the eCLIP-derived peaks (SMI) (Fig. 2b). All ten RBPs also displayed similar binding distributions in the metagene profiles on spliced mRNA features for both ABC and eCLIP (Fig. 2d). Finally, to confirm that ABC and eCLIP were recovering the same

**Fig. 1 | ABC singleplex evaluation. a**, Schematic of ABC and eCLIP workflows. Yellow blocks highlight differences between the two protocols. **b**, Genome browser tracks showing binding sites of RBFOX2 and SLBP for duplicate ABC and eCLIP experiments. Each panel is group-normalized by RPM value. Replicate RBFOX2 data were generated in HEK293XT cells and SLBP data were generated from K562 cells. **c**, Percentage of uniquely mapped reads that are within eCLIP IDR peaks (top) for two replicates. Percent of reads mapping to conserved GCAUG sites (bottom). **d**, Significant peaks of ABC and eCLIP replicate 1 ($P < 0.001$ and greater than eightfold change) in RBFOX2-dependent skipped

exon events, defined as exons alternatively included/excluded upon RBFOX2 shRNA KD[12] (*$P < 0.05$, **$P < 0.001$; ***$P < 10^{-4}$ with two-sided chi-squared test). **e**, Percent of uniquely mapped reads that map to histone mRNAs in eCLIP and ABC libraries, for duplicates, shown separately. **f**, Mean relative information content of reads (IP versus SMI for eCLIP; IP versus RNA-seq for ABC) from ABC or eCLIP across all histone mRNAs. Error bar represents standard error across all histone genes. **g**, Cumulative count of histone genes across the top 100 ranked genes based on two-sided enrichment $P$ value.

binding sites, we computed the overlap coefficient between ABC and eCLIP replicates. There is a notable overlap between enriched peaks in ABC and eCLIP (Fig. 2e). In addition to intra-RBP similarity, there was overlap between RBPs known to bind similar features, like the 5' UTR binding proteins DDX3 and EIF3G. The average read coverage of ABC and eCLIP peaks was also found to be highly correlated for all RBPs (Extended Data Fig. 8).

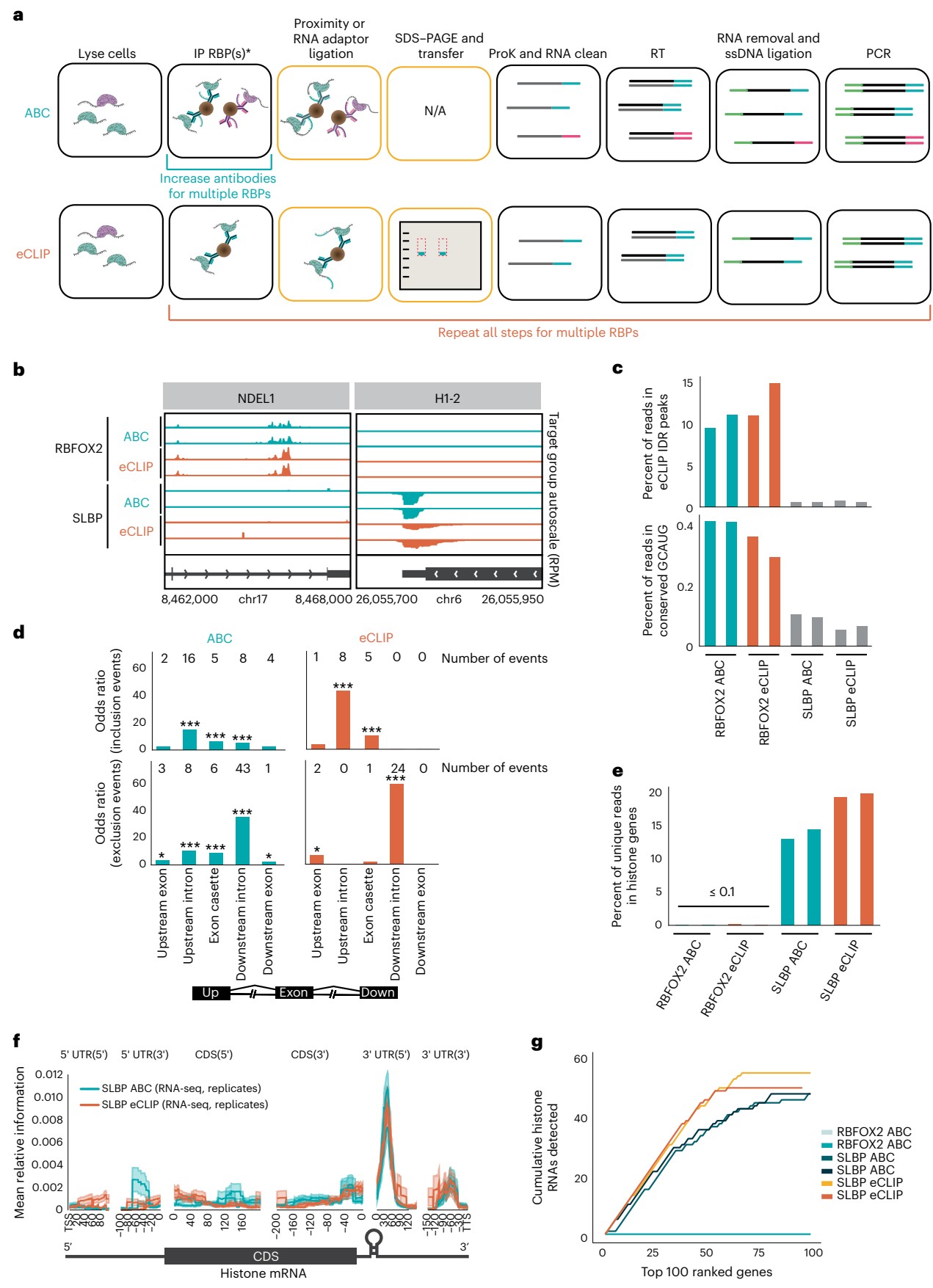

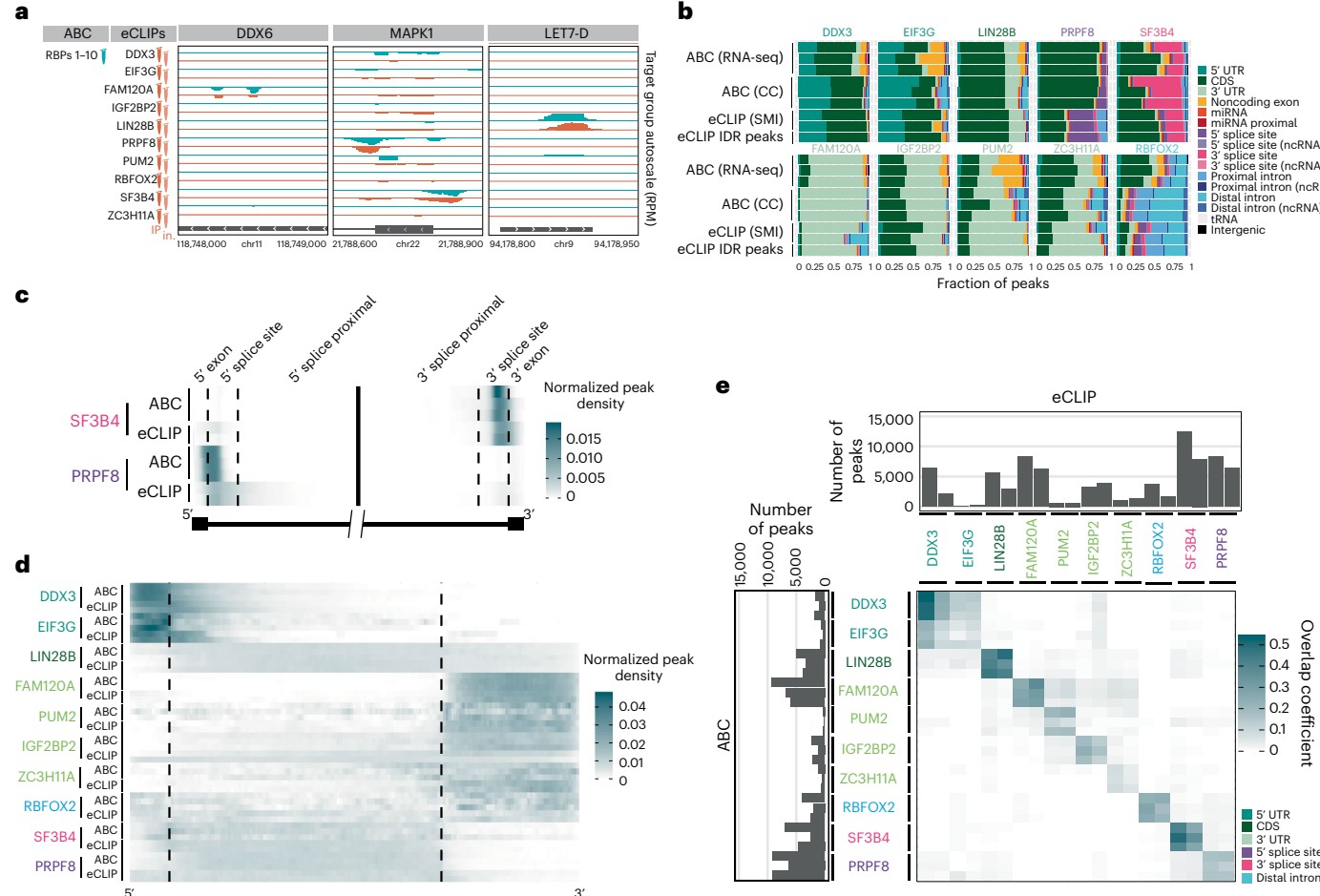

**Fig. 2 | ABC multiplex comparision. a**, Genome browser tracks of selected RBP binding sites depicting similar read coverage between ABC (teal) and ENCODE eCLIP (orange). Each binding site was grouped normalized to all RBPs using RPM. **b**, Stacked bar plots of the fraction of significant peaks (two-sided enrichment $P < 0.001$ and greater than eightfold change) localized to each RNA feature in K562 cells. RBPs are color-coded with their annotated binding feature. Rows are labeled by their respective treatment of 'background' to determine significantly enriched peaks using as background RNA-seq, CC and SMI. Triplicate ABC and duplicate eCLIP datasets are displayed adjacent to each other. eCLIP IDR peaks represent the full dataset downloaded from the ENCODE website. **c**, Density of significant peaks along splicing metagene of the two splicing factors SF3B4 and PRPF8. Peak intensity was normalized such that the total density for each sample

was equal to 1. Duplicate eCLIP and triplicate ABC experiments are displayed adjacent to each other. **d**, ABC enriched peaks (compared with CC) and eCLIP enriched peaks (compared with SMI) were mapped across a merged metagene (protein-coding mRNA) for each RBP. Peak intensity was normalized such that the total density for each sample was equal to 1. Triplicate ABC and duplicate eCLIP datasets are displayed adjacent to each other. **e**, ABC enriched peaks (compared with CC) and eCLIP enriched peaks (compared with SMI) were intersected to find the number of overlapping peaks. The overlap coefficient is defined as (number of overlapping peaks per total number of peaks in the smaller of the two datasets). The total number of peaks are displayed as a bar chart outside of the heatmap. Triplicate ABC and duplicate eCLIP datasets are displayed adjacent to each other.

No differences in peak distributions or quantity were observed when accounting for differences in read depth and peak coverage correlated between single and tenplex (10plex) ABC experiments (Extended Data Fig. 9). This demonstrated that multiplexing RBPs had no appreciable difference on the quality of the data compared with singleplex experiments.

We conclude that a single ABC library (from 1 tube) generates similar overall results to ten separate eCLIP experiments (from 20 tubes). By simply increasing the number of barcodes, this advantage will grow. We note potential limitations. Removal of the SDS–PAGE gel does prevent ABC from resolving higher-order protein–RNA complexes that can be resolved by gel electrophoresis[27]. In multiplexing, the coverage across the different RBPs will be determined by their antibody IP efficiency and expression levels. While using CC to identify peaks that were enriched for specific RBPs within the pool, we obviate the separate SMI library requirement. However, it is important to consider the number and variety of RBPs when implementing CC as these can affect the ranking of protein–RNA interaction sites (Extended Data Fig. 10). As with all CLIP

experiments it is also critical to use high quality antibodies. Despite these caveats, the unprecedented scalability of ABC will facilitate the broad annotation of RBPs in clinically relevant samples, like disease tissues, where source materials are rare and often input-limited.

## Online content

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

## Methods

### Cell culture

K562 (ATCC) and HEK293XT (Takara Bio) cells were cultured in DMEM medium supplemented with 10% fetal bovine serum (FBS) following standard tissue culture technique. Cell pellets were generated by washing 10 cm plates (around 15 million cells) once with 10 ml cold 1× PBS and overlaid or resuspended with minimal (3 ml per 10 cm plate) cold 1× PBS and UV cross-linked (254 nm, 400 mJ cm$^{-2}$) on ice. After crosslinking, cells were scraped and spun down (200$g$ for 3 min), the supernatant removed, and washed with 10 ml cold 1× PBS. Cell pellets (10 million each) were flash-frozen on dry ice and stored at −80 °C.

### Oligo barcoding prep

A 100 µl sample of 100 µM oligo barcode (IDT) in PBS and 10 µl 10 mM azide-NHS (Click Chemistry Tools catalog no. 1251–5) in DMSO were rotated at room temperature for 2 h. Unreacted azide was removed by buffer exchanging into PBS using Zeba desalting columns (Thermo catalog no. 89883) following the manufacturer's protocol. Azide labeled barcodes were stored at −20 °C.

### Antibody barcoding

Antibodies (20 µg) were diluted to 70 µl in PBS (enough for four IPs), and the buffer was exchanged into PBS using Zeba desalting columns following the manufacturer's protocol. Then, 10 µl 10 mM DBCO-NHS (Click Chemistry Tools catalog no. A134-10) was added to the antibodies and allowed to rotate at room temperature for 1 h[28]. Unreacted DBCO-NHS was removed by buffer exchanging into PBS using Zeba desalting columns and stored at 4 °C.

Azide (6.65 µl) containing barcodes was then reacted with all the DBCO labeled antibodies (around 70 µl). Make sure to note which barcode is attached to each antibody. The mixture was allowed to rotate overnight at room temperature. Labeled antibodies were stored at 4 °C and assumed to be 20 µg and used as is.

Antibodies and oligonucleotides used in this study can be found in Supplementary Table 3.

### Antibody conjugation CLIP

**IP bead conjugation.** Lysis buffer (200 µl; 50 mM Tris pH 7.4, 100 mM NaCl, 1% Igepal, 0.1% SDS, 0.5% sodium deoxycholate) was added to 25 µl anti-rabbit Dynabeads (Thermo Fisher catalog no. 11204D). Beads were washed twice with 500 µl lysis buffer before being resuspended in 50 µl lysis buffer. Antibody (5 µg; around 20 µl from antibody barcoding step) was added and rotated at room temperature for 1 h. Beads were again washed twice with 500 µl lysis buffer and resuspended in 50 µl lysis buffer. The procedure was repeated for each barcode and antibody combination.

**Immunoprecipitation.** Frozen HEK293XT or K562 cell pellets (around 10 million cells) were lysed in 1 ml lysis buffer supplemented with 5 µl protease inhibitor cocktail (Thermo Fisher catalog no. 87786) and 10 µl RNase inhibitor (NEB catalog no. M0314B) and sonicated for 5 min with 30 s on/off cycles at 4 °C. The lysate was then treated with 10 µl diluted (1:25) Rnase I (Thermo Fisher catalog no. AM2295) and 5 µl TurboDNase (Thermo Fisher catalog no. AM2239B001) and mixed (1,200 rpm) at 37 °C for 5 min. Cellular debris was removed by centrifugation at 12,000$g$ for 3 min. The supernatant was then transferred to a new tube along with 50 µl of each preconjugated antibody for each barcode (50 µl each from ten different barcoded antibodies, 500 µl total for 10plex) coated magnetic beads and immunoprecipitated overnight by rotation at 4 °C. Beads were subsequently washed with 500 µl high salt wash buffer (50 mM Tris pH 7.4, 1 M NaCl, 500 mM EDTA, 0.5% Igepal, 1% SDS, 0.5% sodium deoxycholate) (3×), high salt wash buffer + 80 mM LiCl (1×) and low salt wash buffer (500 mM Tris pH 7.4, 250 mM MgCl$_2$, 5% Tween 20, 125 mM NaCl) (3×).

**Proximity ligation.** Beads were resuspended in 76 µl T4 PNK reaction buffer (97.2 mM Tris pH 7, 13.9 mM MgCl$_2$, 1 mM ATP), 3 µl T4 PNK (NEB catalog no. M0201B), 1 µl Rnase inhibitor, and incubated at 37 °C for 20 min with interval mixing (1,200 rpm). After PNK treatment, samples were washed with 500 µl high salt buffer (1×) followed by low salt buffer (3×). Proximity barcode ligations were carried out in 150 µl T4 ligation reaction mix (11 µl T4 ligase (NEB catalog no. M0437B-BM), with 94 µl ligation buffer (75 mM Tris pH 7.5, 16.7 mM MgCl$_2$, 5% DMSO, 0.00067% Tween 20, 1.67 mM ATP, 25.7% PEG8000), 2 µl RNase inhibitor and 43 µl water) at room temperature for 45 min with interval mixing (1,200 rpm). Samples were again washed with high salt buffer (1×) and low salt buffer (2×). Chimeric RNA barcode molecules were eluted from the bead by incubating with 127 µl ProK digestion solution (17 µl ProK (NEB catalog no. P8107B), 110 µl (100 mM Tris pH 7.5, 50 mM NaCl, 10 mM EDTA, 0.2% SDS)) at 37 °C for 20 min followed by 50 °C for 20 min with interval mixing (1,200 rpm). Samples were placed on the Dynamagnet and supernatants were transferred to a clean tube. Samples were cleaned up with Zymogen RNA clean and concentrator following manufacturers protocol and eluted in 10 µl.

**Reverse transcriptase and library prep.** RNA (9 µl) was combined with 1.5 µl reverse transcriptase (RT) primer mix (100 µl 10 mM dNTPs, 10 µl 10 µM ABC RT primer) and was heated to 65 °C for 2 min and immediately placed on ice. Then, 9.2 µl of RT buffer (2.17× SuperScript III RT buffer, 10 mM DTT), 0.2 µl RNase inhibitor, and 0.6 µl of Superscript III was added to the sample, mixed by pipetting and reverse transcribed at 54 °C for 20 min. After RT, excess primers and nucleotides were removed with 2.5 µl ExoSAP-IT (Thermo Fisher catalog no. 75001.10.ML) at 37 °C for 15 min and RNA was degraded with the addition of 1 µl 0.5 M EDTA and 3 µl 1 M NaOH and heated at 70 °C for 10 min. The sample was pH neutralized with 3 µl 1 M HCl.

**RT cleanup.** MyOne Silane beads were prepared by adding 5 µl beads to a fresh tube containing 25 µl RLT buffer (Qiagen cat# 79216) + 0.01% Tween 20. The tube was placed on a magnet and supernatant removed and replaced with 93 µl RLT buffer plus 0.01% Tween 20; 90 µl bead preparation was then added to the pH neutralized RT cDNA and incubated at room temperature for 10 min. Beads were washed with 300 µl of 80% ethanol twice. Following the last wash the beads were allowed to air dry until they no longer appeared wet. The cDNA was then eluted in 2.5 µl ssDNA ligation adapter (50 µl 100 µM ABC i7 primer, 60 µl DMSO, 140 µl Bead Elution Buffer) and heated to 70 °C for 2 min before being placed on ice.

**ssDNA ligation.** Without removing the beads, 6.5 µl T4 ligase solution (76.9 mM Tris pH 7.5, 15.4 mM MgCl$_2$, 3% DMSO, 30.8 mM DTT, 0.06% Tween 20, 1.5 mM ATP, 27.7% PEG8000), 1 µl T4 ligase, and 0.3 µl deadenylase (NEB catalog no. M0331B) was added and rotated overnight at room temperature. Bead binding buffer (45 µl; 0.001% Tween 20, 10 mM Tris pH 7.5, 0.1 mM EDTA) and 45 µl ethanol were added to the ligation mixture to rebind the cDNA to the silane beads for 10 min at room temperature. Beads were washed with 300 µl of 80% ethanol twice. Following the last wash, the beads were allowed to air dry until the beads no longer appeared wet. cDNA was eluted in 25 µl elution buffer (0.001% Tween 20, 10 mM Tris pH 7.5, 0.1 mM EDTA)

### PCR quantification

cDNA (1 µl) was diluted with 10 µl water; 1 µl of the diluted cDNA was mixed with 2 µl of each qPCR primer (1.25 µM in water) and 5 µl Luna qPCR Master Mix. Samples were processed on a StepOnePlus System. Final libraries were amplified with dual index Illumina primers using Next Ultra II Q5 Master Mix. If necessary to remove adapter dimers, libraries were run and extracted from a 1–2% eGel using a Qiagen Gel Extraction kit. Library were quantified by Tapestation and sequenced on an Illumina Nextseq 2000. Following eCLIP guidelines, libraries were

sequenced at around 25 million reads per barcode (that is, 250 million reads for a 10plex).

## Data preprocessing

Data were processed similarly to the standard eCLIP pipeline[12], except for a few adjustments to the multiplex design and library structure of ABC. For ABC data, unique molecular identifiers (UMIs) were extracted using umitools v.1.0.0 (ref. [29]), and adapters were trimmed twice using cutadapt v.2.8 (ref. [30]). Fastqs files were demultiplexed based on the 5′ nucleotide barcode sequence using fastx toolkit (http://hannonlab.cshl.edu/fastx_toolkit/). ABC libraries were sequenced on the reverse strand. Therefore, reads were reverse complemented before alignment to repetitive regions, removal of multi-mapped reads and alignment to the genomic sequences using STAR v.2.7.6 (ref. [31]). The pipeline is available at https://github.com/YeoLab/oligoCLIP.git.

## Calculating enrichment of peaks across different background inputs

We used CLIPper v.2.1.2 (https://github.com/YeoLab/clipper) to identify peaks from the IP library[23]. The number of ABC/eCLIP reads overlapping CLIPper-identified peaks and the number overlapping the identical genomic region in the 'background' sample, were counted and used to calculate fold enrichment (normalized by total usable read counts in each dataset), with enrichment $P$ value calculated by Yates' chi-squared test (Perl) (or Fisher's exact test (calculated in the R statistical computing software) where the observed or expected read number was below five), which have minimal reportable $P$ values of $10^{-88}$ (for chi-squared) and $2.2 \times 10^{-16}$ (for Fisher's exact test), respectively. We evaluated different backgrounds: SMI control from eCLIP experiments, RNA-seq data[25,26] (HEK293: GSE122425, K562: ENCODE project ENCSR000AEL) and CC from multiplexed ABC.

To compute our enrichment, we denote the number of reads in a region in the IP library, the number of reads in the region in the background library (can either be RNA-seq, SMI or CC), the total number of uniquely mapped reads in the IP library and the total number of reads in the background. For the region, we create a contingency table of the form (Inside peak i, Outside peak) × (In IP, In Background),

This script is implemented in scripts/overlap_peakfi_with_bam.pl and is wrapped around in rules/chi.py (for using the sum of all other multiplexed libraries) and rules/snakeCLIPper.py (for using another library, either RNA-seq or SMI). The pipeline is available at https://github.com/YeoLab/oligoCLIP.

## Peak filtering strategy

To ensure removal of all repetitive elements we performed a reverse intersection of all peak files with the repeatmasker bed file downloaded from the University of California Santa Cruz (UCSC) table viewer[32]. For Fig. 1g, highly abundant background RNAs (mitochondrial and snoRNAs) were removed.

## RBFOX2 peak splicing analysis

Since no large-scale gold-standard standard datasets of binding sites exists, we made assumptions based on previous knowledge of certain RBPs. RBFOX2 is also known to be enriched proximal to its regulated exons, which exhibit positional dependent alternative splicing (Fig. 1d)[22]. We utilized the splicing microarray-defined ($n = 150$) differentially included and skipped cassette exon events upon loss of RBFOX2 (ref. [12]). The genome coordinates are subsequently converted to GRCh38 using UCSC liftover. We define the following regions surrounding each cassette exon: the upstream exon is defined as the exon 5′ to the cassette exon; the downstream exon is defined as the exon 3′ to the cassette exon and the upstream intron is defined as the intron between the upstream exon and the cassette exon. The downstream flanking intron is the intron between the cassette and downstream exon. To test if there is enrichment of significant peaks ($P < 0.001$ and were greater than eightfold change) in each region, we used the chi-squared test to test for significance in enrichment in binding in the above defined region, against a set of randomly sampled cassette exons ($n = 1,500$, 'background events') with no change upon RBFOX2 KD. Any peak 'overlaps' with a region if at least 50% of peak length falls into the designated region. Odds ratio was calculated as: ((the number of skipped exons that overlapped with significant peak)/(the number of skipped exons that do not overlap with significant peak))/((the number of exons that overlap with background exons)/(the number of exons that do not overlap with background exons)).

## RBFOX2 conserved motif analysis

RBFOX2 has a strong binding to the GCAUG motif and its sites exhibit high sequence conservation across vertebrate evolution (which we define operationally as GCAUG sequences with phyloP[33] greater than three, in intronic and UTR regions).

## Region annotation

To understand which transcriptomic feature each RBP tends to bind, we use GENCODE v.35 annotations[34], and an inhouse pipeline (https://github.com/YeoLab/annotator) to prioritize region when a peak overlaps with several regions (Extended Data Figs. 3a and 4a and Fig. 2). Proximal intron is defined as 500 base pairs to the splice site. Splice sites were defined as within 100 base pairs of the annotated splice site.

## Motif analysis

We ran motif analysis on significant peaks ($P < 0.001$ and greater than eightfold change) using an inhouse pipeline (https://github.com/YeoLab/clip_analysis_legacy/tree/889df77bbbd23679833a0744d3aa29b3f6bcb6d9). Briefly, it wraps around HOMER[27]. For peaks in each region (UTR, CDS, and so on), a set of background regions of matched size and GC content is generated. HOMER is then deployed comparing the sequence in peaks versus the background in search of a significant motif.

## Metagene analysis

Here, we presented two types of metagene analysis, one with raw reads (Fig. 1f), the other with significant peaks (Fig. 2c,d). For the raw read metagenes (Fig. 1f) we used a software called Metadensity (https://github.com/YeoLab/Metadensity, manuscript submitted) that calculates the relative information content (RIC) of immunoprecipitated reads versus the background (SMI/RNA-seq). RIC here serves as an approximation of binding distribution in the transcript. For each position in a transcript, the fraction of reads truncated at each position is compared with the fraction of truncation in background. For Fig. 1f, we calculated RIC for every histone transcript, then averaged the results.

## Comparing peaks between ABC and eCLIP

BedTools was utilized to calculate the 'overlap coefficient' of two sets of significant peaks, defined as (number of overlapping peaks per total number of peaks in the smaller of the two datasets)[35] (Fig. 2e).

## Comparing target ranking between methods and backgrounds

To determine the top targets for SLBP we first ranked all peaks by their $P$ value and then dropped duplicate gene names (Fig. 1g and Extended Data Fig. 7). Since SLBP is known to bind histone RNAs, we then assigned a value of 1 instead of 0 to all histone gene names appearing in the dataset[28]. A similar strategy was used to annotate the ten RBPs in the multiplex but instead of gene name labels we used RNA feature labels and assigned values only to the correct feature for each RBP.

## Comparing background rate of SMI peak removal

Peaks from eCLIP datasets that did not pass our thresholds ($P < 0.001$ and with greater than eightfold change) after accounting for SMI reads

are probably nonspecific interactions, termed 'experimental background.' To evaluate whether our normalization approaches for ABC removed nonspecific peaks, we can compare them with this background (Extended Data Fig. 7).

### Estimating library complexity

Library complexity, that is, the number of unique molecules captured in the experiment, is a function of efficiency of every step within the library preparation as well as the sequencing depth. To ensure ABC has the same efficiency in capturing uniquely bound RNAs, we estimated at various sequencing depths of uniquely mapped reads, how many UMIs can be captured (Extended Data Fig. 2). Uniquely mapped reads were downsampled to various depths, then followed the preprocessing pipeline to deduplicate the reads.

### Reporting summary

Further information on research design is available in the Nature Portfolio Reporting Summary linked to this article.

### Data availability

Data are available at GEO accession code GSE205536. ENCODE eCLIP IP, size-matched inputs and RNA-seq were downloaded from https://www.encodeproject.org/.

### Code availability

All code for analysis is accessible at https://github.com/algaebrown/oligoCLIP.git.

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

### Acknowledgements

We thank J. Dessert for creating the schematic artwork. G.W.Y. is supported by National Institutes of Health grants (nos. HG004659 and HG009889) and by an Allen Distinguished Investigator Award—a Paul G. Allen Frontiers Group advised grant of the Paul G. Allen Foundation.

### Author contributions

ABC was invented and initial proof-of-concept experiments and analysis were performed by D.A.L. and K.B.C. Subsequent experiments were performed by D.A.L., K.R. and A.C.N. with bioinformatic analysis by D.A.L., K.A.S., H.-L.H., K.R.H., S.C.B., S.A.M., and B.A.Y. H.-L.H. and B.A.Y. developed the github code repository for ABC analysis. The manuscript was written by D.A.L., H.-L.H., K.R. and G.W.Y. with input from all authors.

### Competing interests

D.A.L. and K.B.C. are listed as authors on a patent application related to this work. D.A.L., K.A.S., K.R.H., S.C.B., S.A.M., A.C.N. and K.B.C. are paid employees of Eclipse Bioinnovations. G.W.Y. is a cofounder, member of the Board of Directors, on the SAB, equity holder and paid consultant for Locanabio and Eclipse Bioinnovations. G.W.Y. is a visiting professor at the National University of Singapore. G.W.Y.'s interests have been reviewed and approved by the University of California, San Diego, in accordance with its conflict-of-interest policies. The authors declare no other competing interests.

### Additional information

**Extended data** is available for this paper at https://doi.org/10.1038/s41592-022-01708-8.

**Correspondence and requests for materials** should be addressed to Karen B. Chapman or Gene W. Yeo.

A.

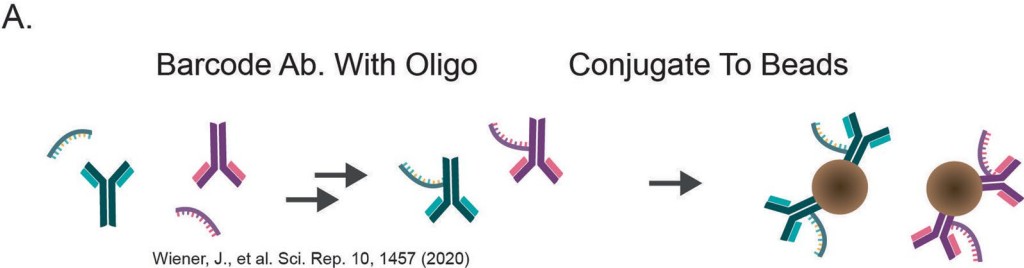

Barcode Ab. With Oligo    Conjugate To Beads

Wiener, J., et al. Sci. Rep. 10, 1457 (2020)

B.

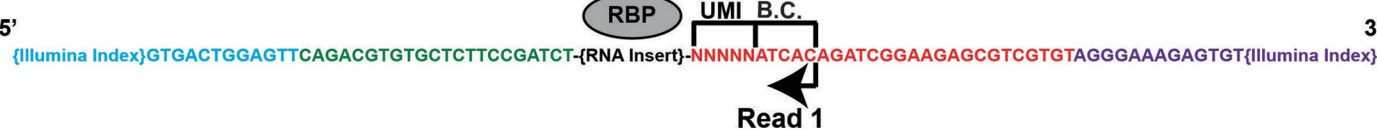

**Extended Data Fig. 1 | Antibody barcode scheme.** A) Schematic of antibody barcoding required before immunoprecipitation. B) ABC read structure. The UMI was designed to be between the barcode sequence and RNA insert to avoid any ligation bias due to defined barcode sequences.

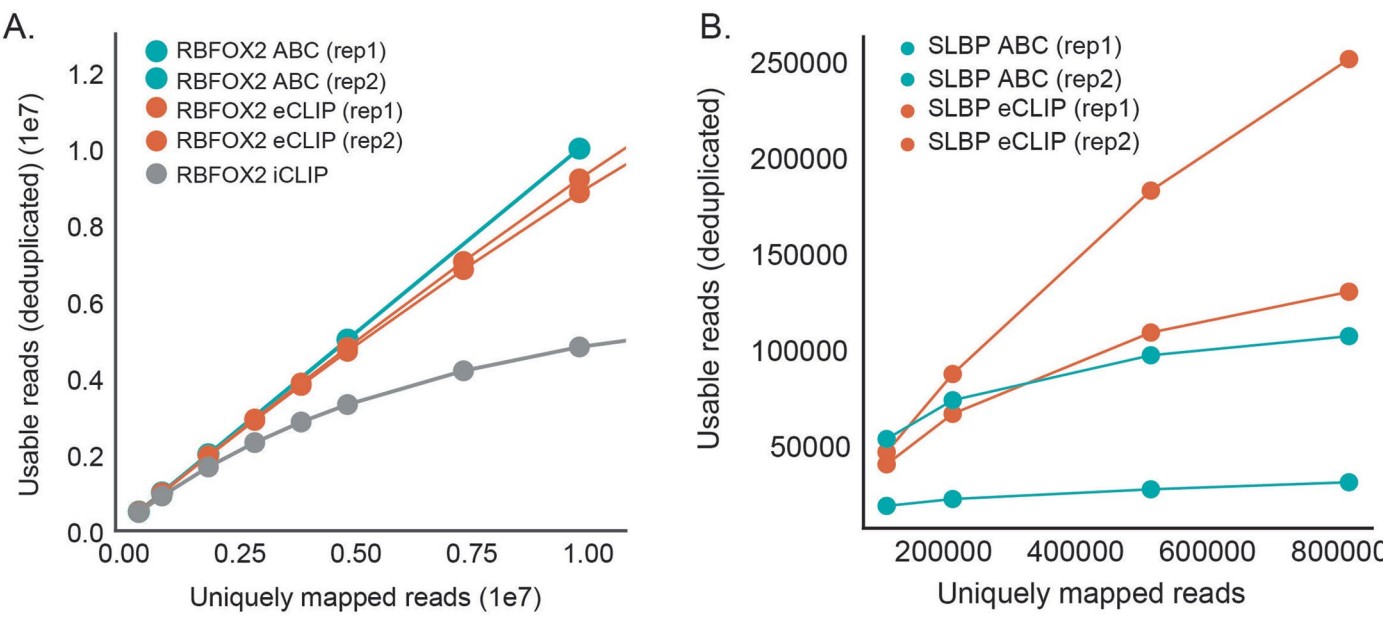

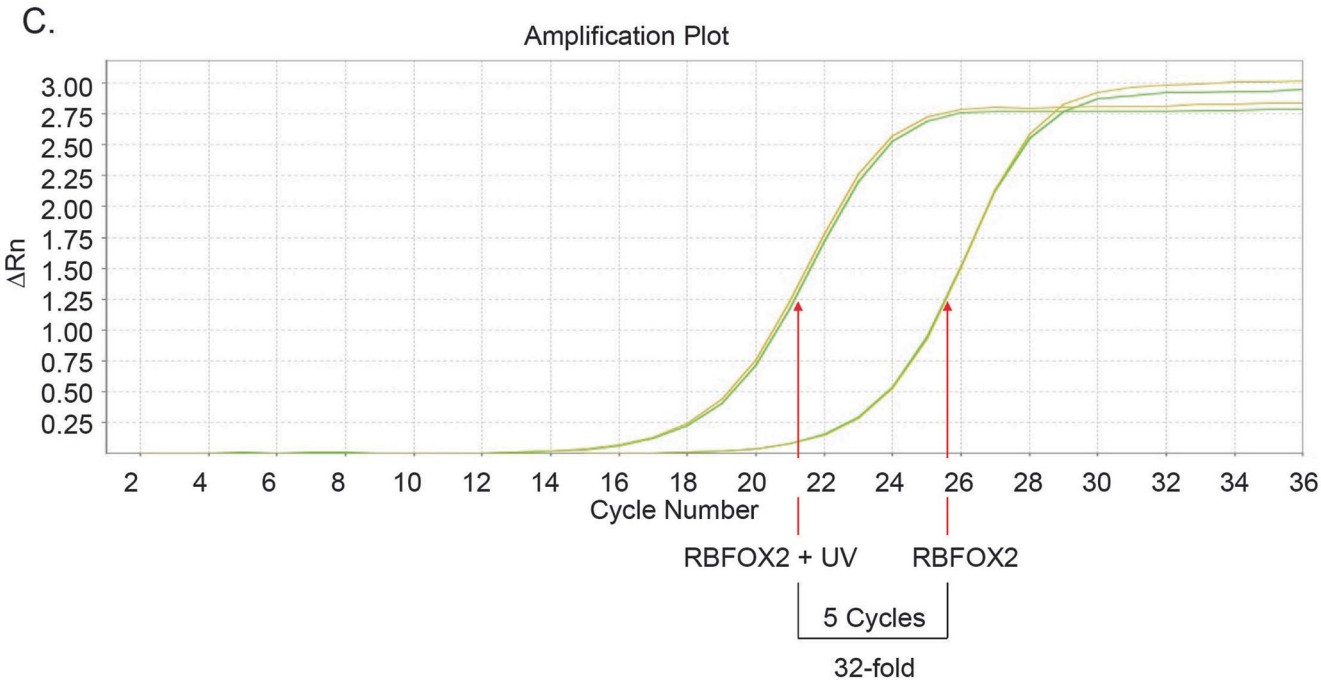

**Extended Data Fig. 2 | Singleplex library complexity.** The number of unique molecules was estimated as a function of sequencing depth. After randomly sampling uniquely mapped reads from ABC, eCLIP, and iCLIP, we plot the number of uniquely mapped reads vs the number of usable reads. A) RBFOX2 (HEK293XT) B) SLBP (K562). C) qPCR amplification plot of RBFOX2 ABC with and without UV crosslinking.

A.

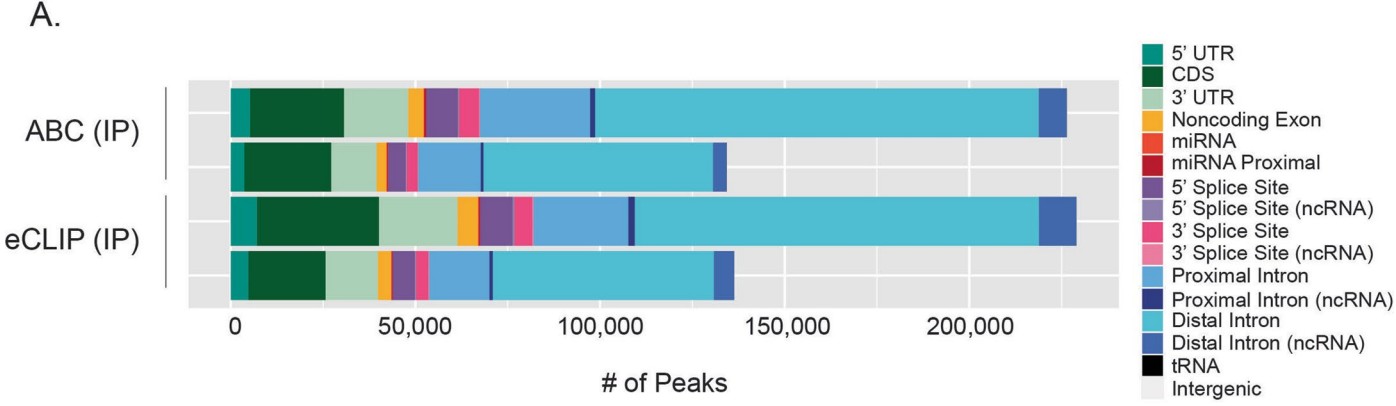

B.

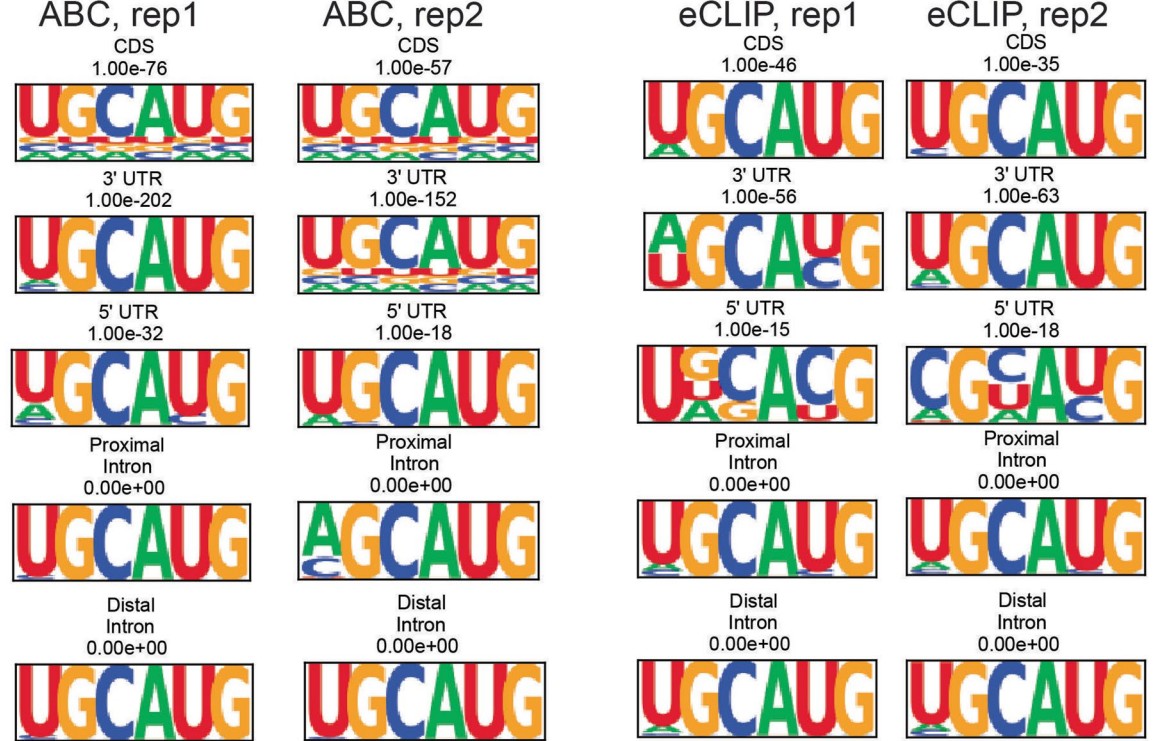

**Extended Data Fig. 3 | RBFOX2 singleplex.** A) Stacked bar plots of the total number of peaks localized to each RNA feature from HEK293XT cells. B) HOMER motif analysis of the significant peaks (two-sided enrichment *P* < 0.001 and >8-fold change). Peaks were stratified by region (CDS, 3'UTR, proximal intron, or distal intron).

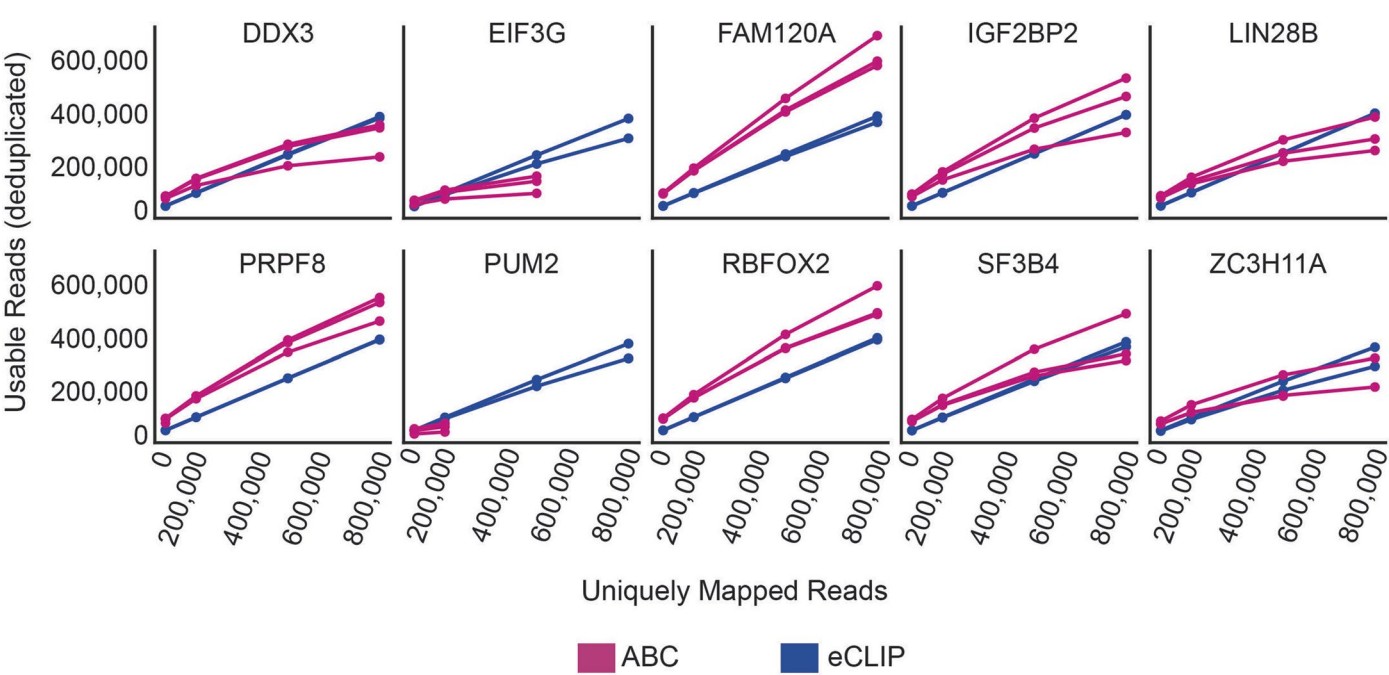

**Extended Data Fig. 4 | Multiplex library complexity.** The number of unique molecules was estimated as a function of sequencing depth. After randomly sampling uniquely mapped reads from ABC and eCLIP, we plot the number of uniquely mapped reads relative to the number of usable reads across the 10 RBPs within the multiplex.

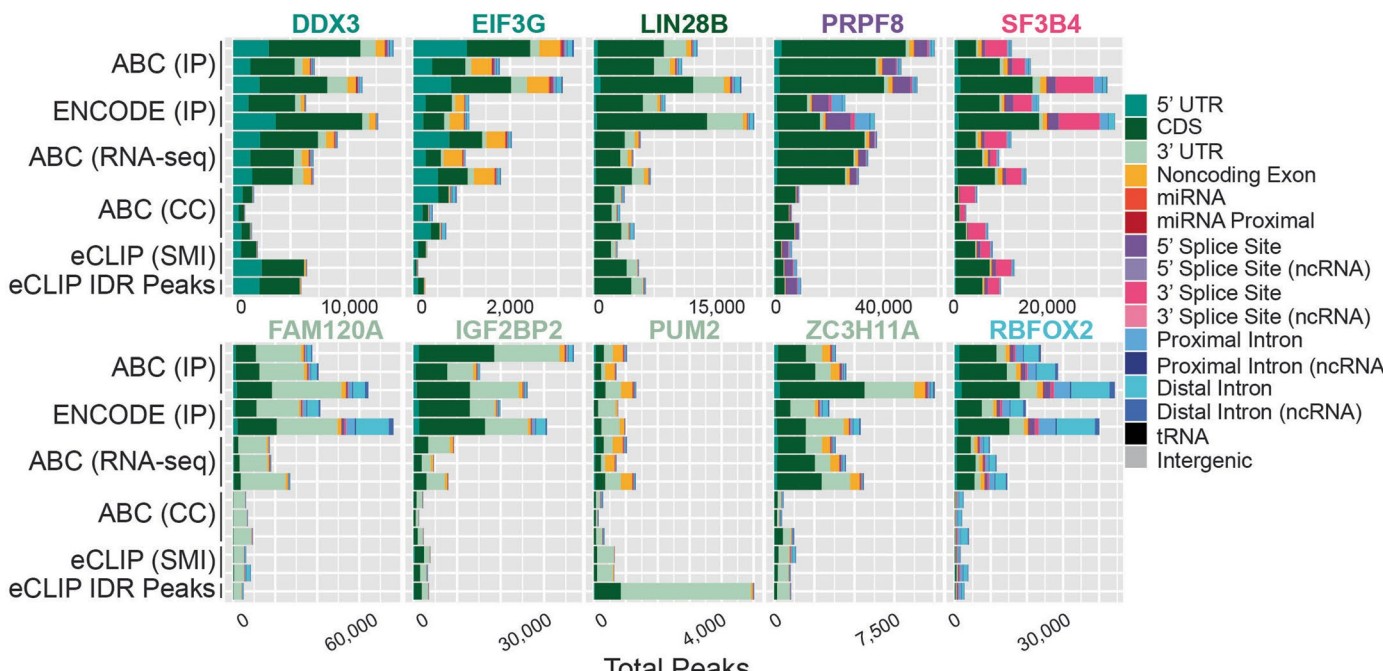

**Extended Data Fig. 5 | Total peak counts.** Stacked bar plots of the total number of peaks localized to each RNA feature in K562 cells. RBPs are color coded with their annotated binding feature. Rows are labeled by their respective normalization. Triplicate ABC and duplicate eCLIP datasets are displayed adjacent to each other. IDR peaks were not downsampled and used as is from ENCODE.

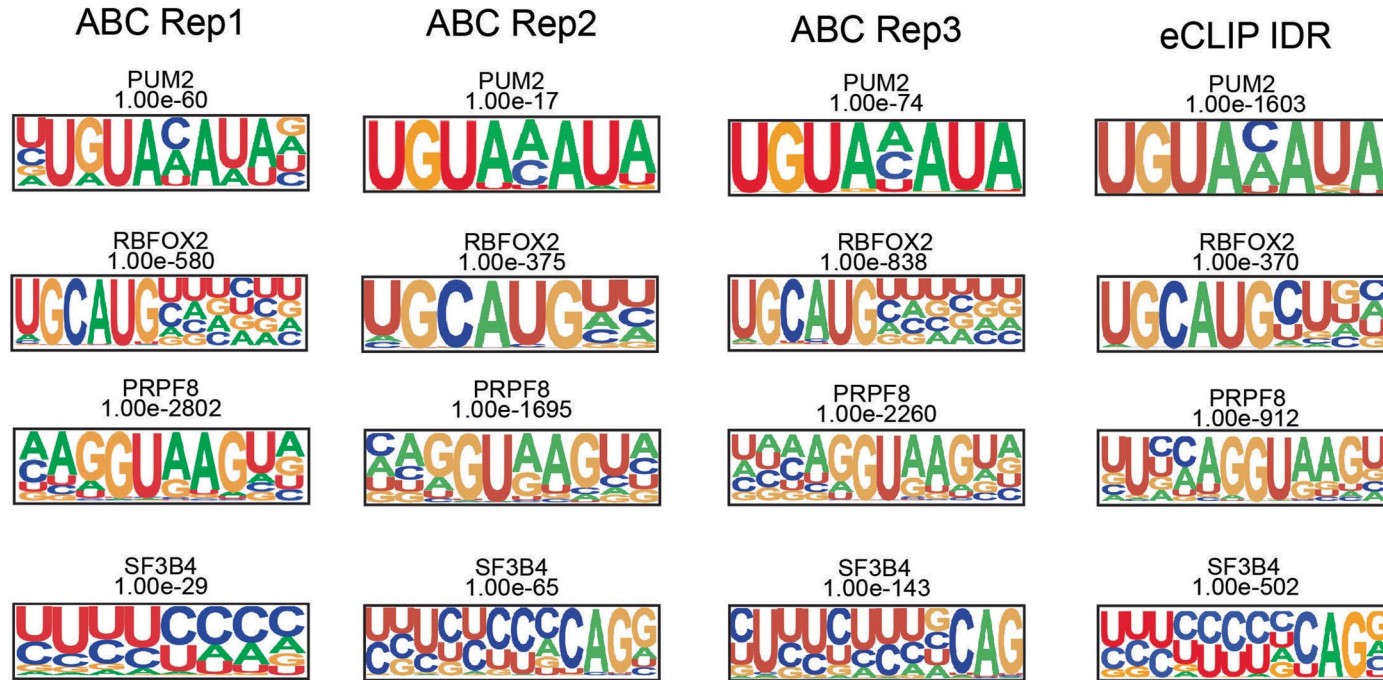

**Extended Data Fig. 6 | HOMER de novo motifs.** De novo motifs detected for PUM2, RBFOX2, PRPF8, and SF3B4 called from ABC CC peaks and eCLIP IDR peaks (total ENCODE). Two-sided enrichment *P* values are listed for each RBP and sample.

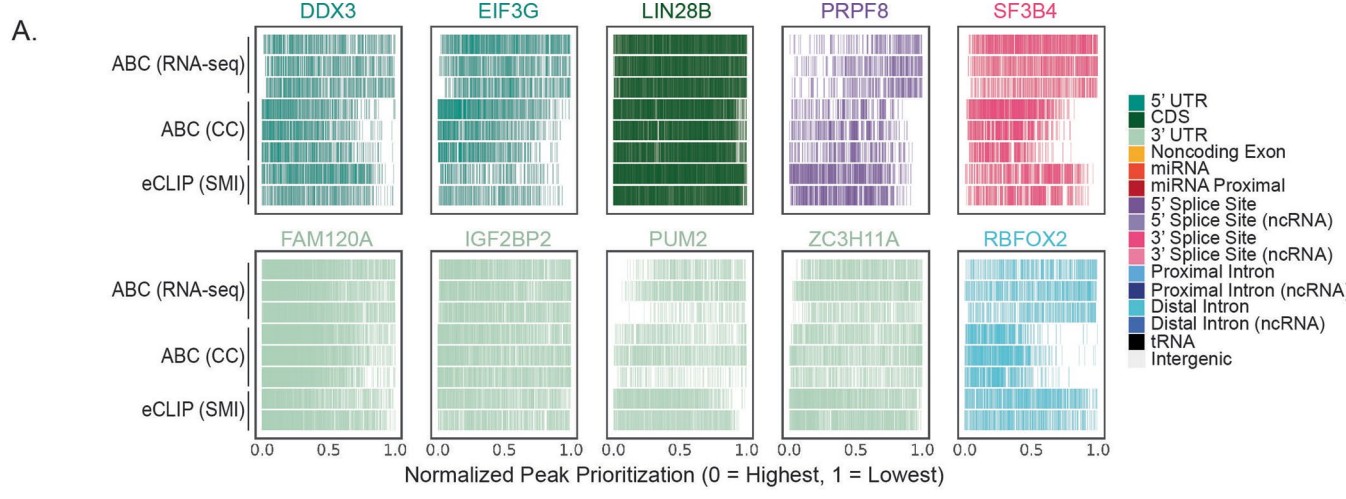

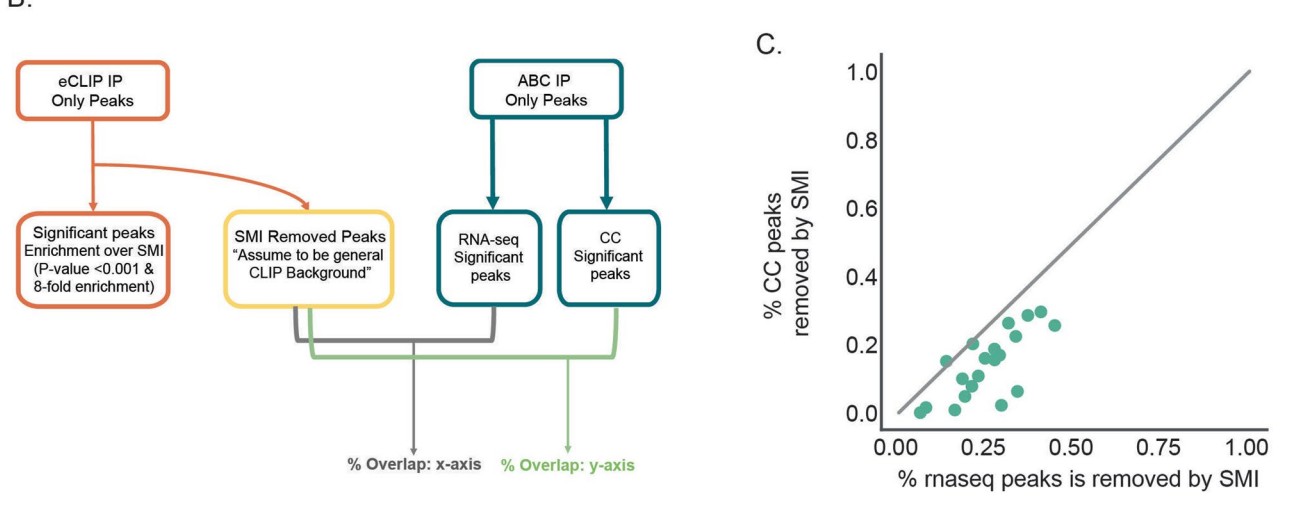

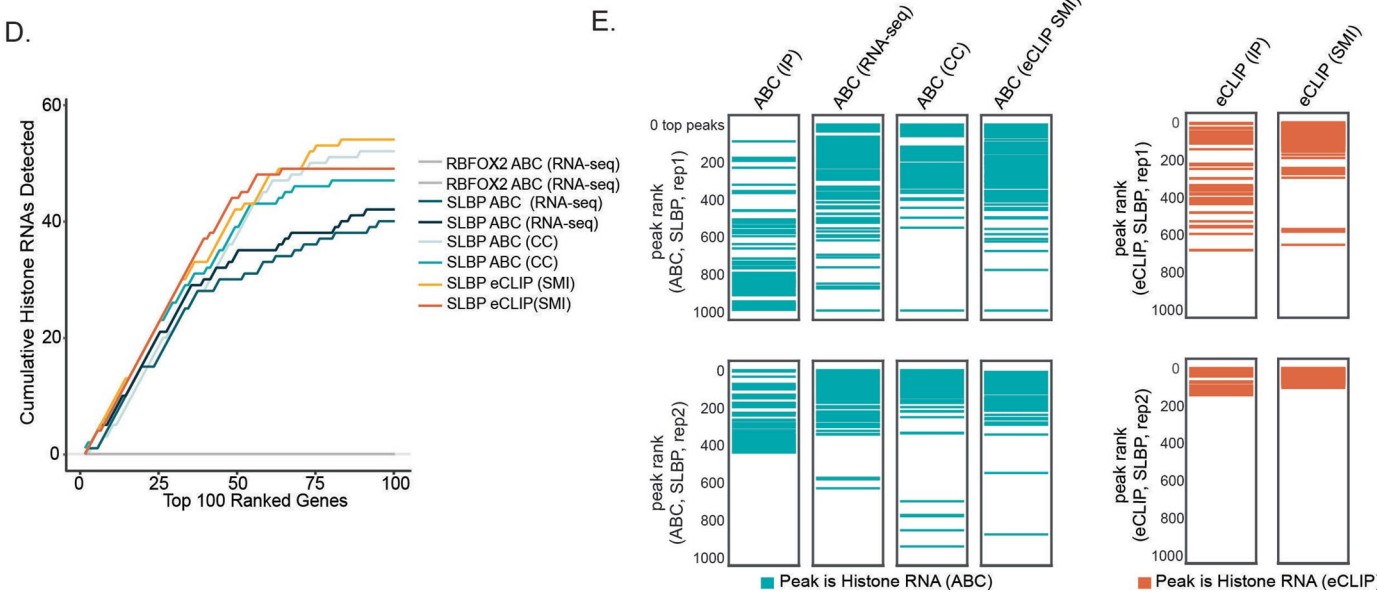

**Extended Data Fig. 7 | See next page for caption.**

**Extended Data Fig. 7 | Complement control peak prioritization.** A) Sorted bar plots ranked by two-sided enrichment peak p-values. Colors correspond to peaks annotated as matching known RBP function. Triplicate ABC and duplicate eCLIP datasets are displayed adjacent to each other. B) Schematic of SMI overlap with ABC peaks. C) Scatter plot of the percent of peaks removed by SMI compared to RNA-seq and CC normalizations for each barcode within the multiplex (rep 2 and 3). D) Cumulative count of histone genes across the top 100 ranked genes based on two-sided enrichment p-values using different methods of normalization. E) Sorted bar plots of ranked peaks by one-sided enrichment p-value < 0.001. Colors denote that the peak is within an annotated histone RNA.

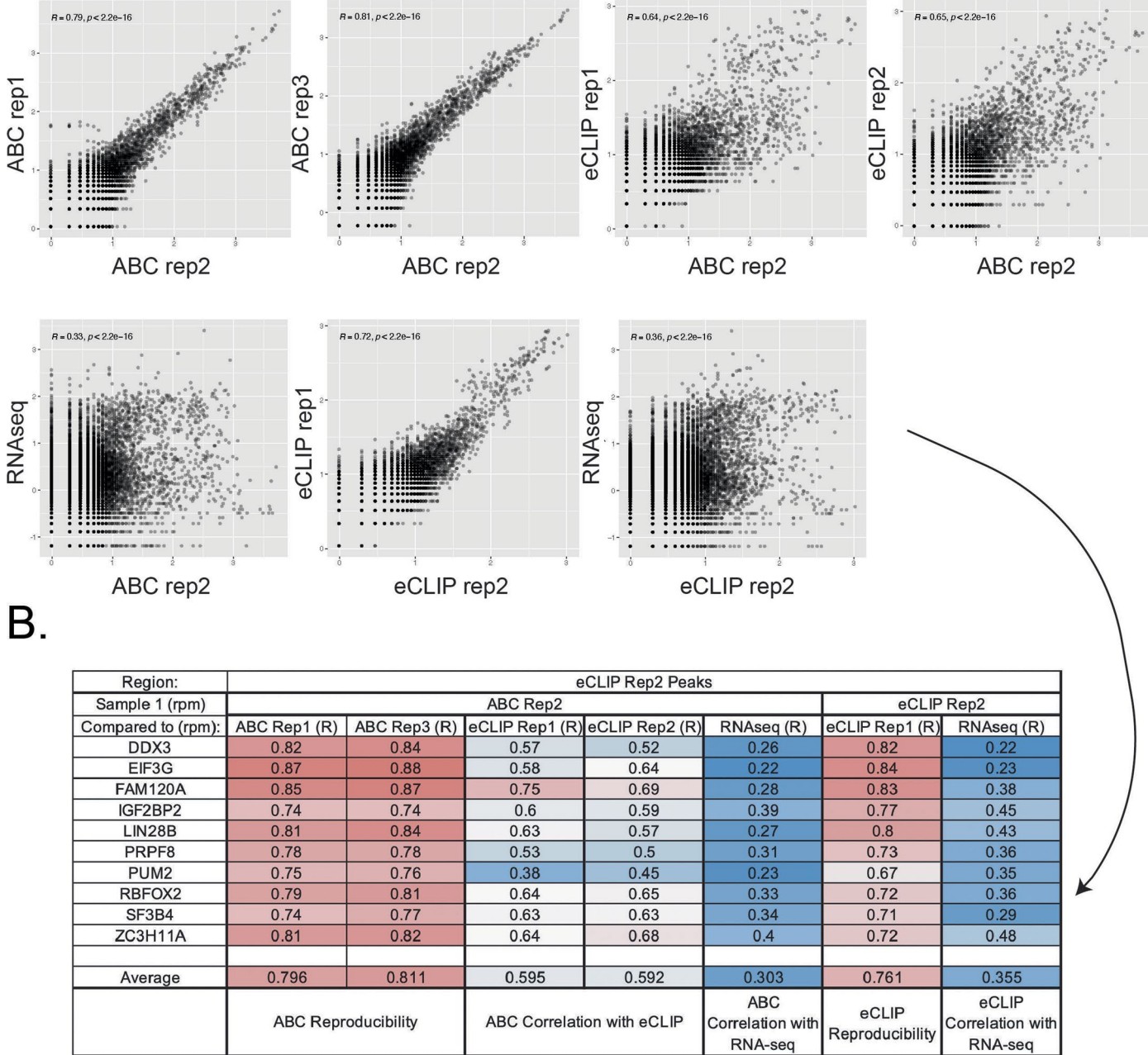

**Extended Data Fig. 8 | ABC and eCLIP correlations.** A) Examples of scatter dot plots of reads (RPM) within RBFOX2 eCLIP-defined binding sites between eCLIP, ABC and RNA-seq. P-values are two-sided. B) Table of Pearson correlation values summarizing these comparisons for the different RBPs.

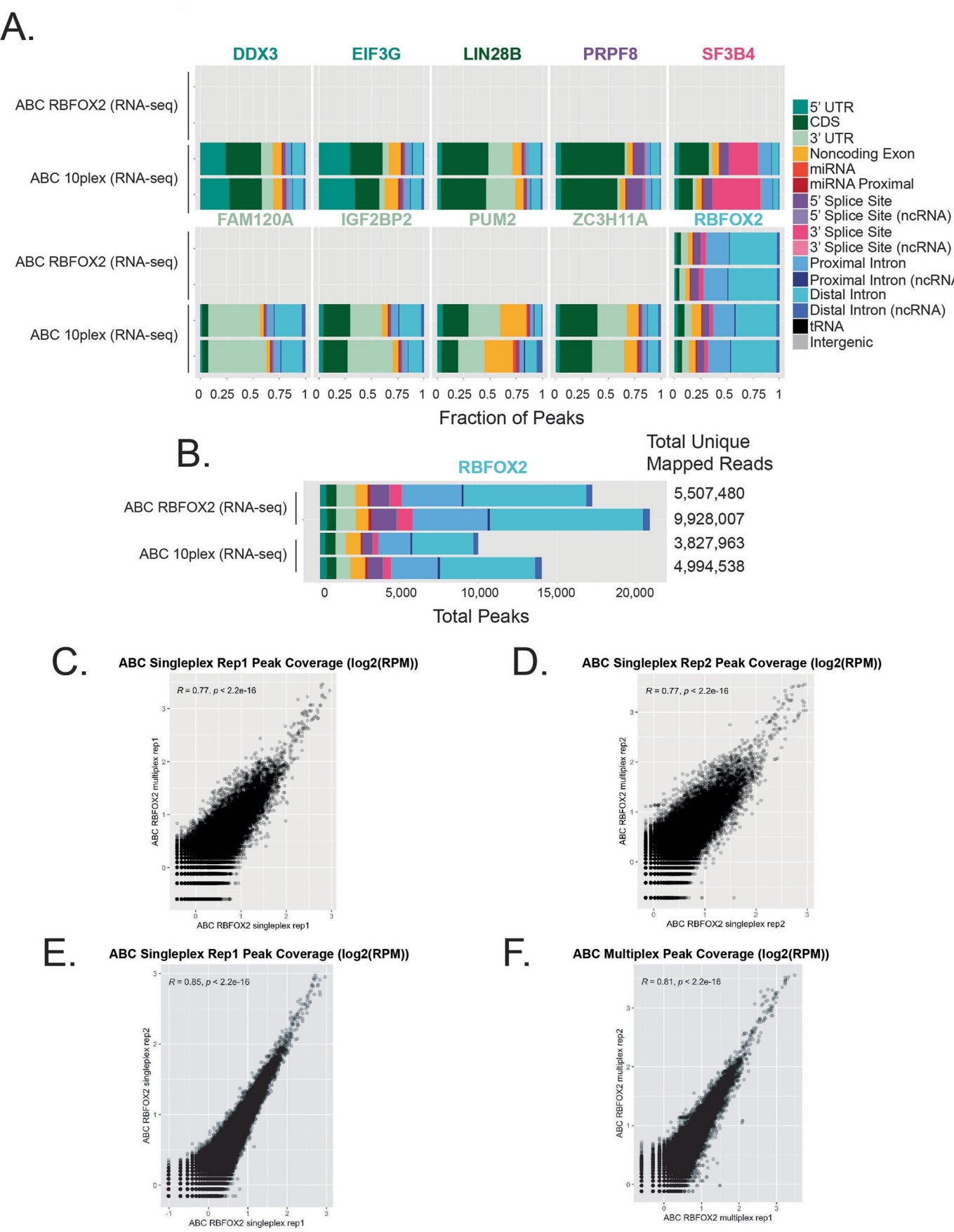

**Extended Data Fig. 9 | See next page for caption.**

**Extended Data Fig. 9 | RBFOX2 singleplex vs multiplex.** A) Stacked bar plots of the fraction of peaks localized to each RNA feature comparing single plex and multiplex ABC in HEK239XT cells. B) Total number of peaks detected in single plex and multiplex ABC in HEK239XT cells with total uniquely mapped reads listed on the right. C) Correlation of peak coverage between replicate 1 of singleplex ABC vs multiplex ABC for RBFOX2. D) Correlation of peak coverage between replicate 2 of singleplex ABC vs multiplex ABC for RBFOX2. E) Correlation of peak coverage between duplicate ABC singleplex RBFOX2 experiments. F) Correlation of peak coverage between duplicate ABC multiplex RBFOX2 experiments. All displayed p-values are two sided.

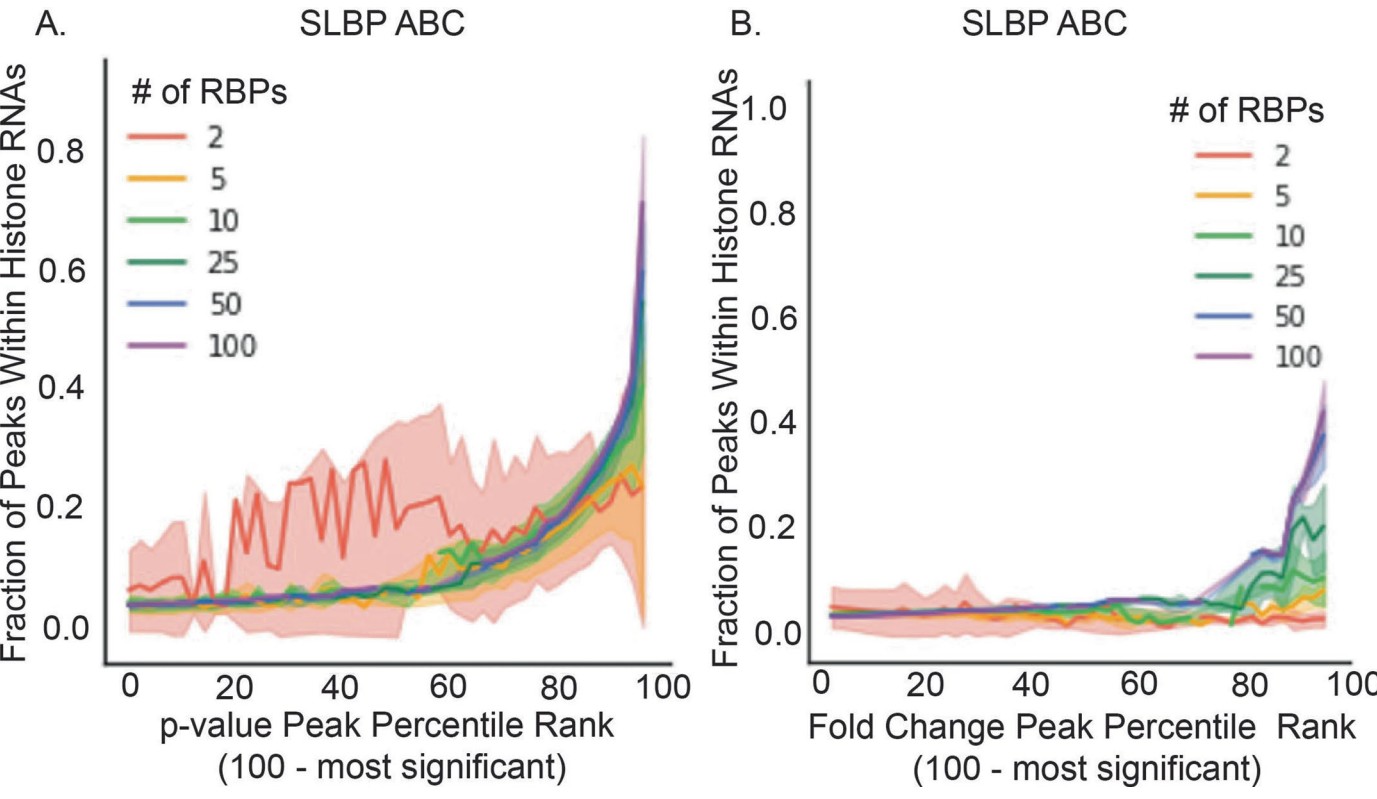

**Extended Data Fig. 10 | Evaluating the number of RBPs applied to SLBP complement control.** Fraction of peaks within histone mRNAs for SLBP single-plex ABC observed based on various numbers of other ENCODE K562 RBPs used in CC analysis. A) Ranked by p-value. B) Ranked by fold enrichment. Error bar represents the standard deviation across 6 random permutations of selected eCLIP datasets as CC.

Corresponding author(s): Daniel A. Lorenz, Hsuan-Lin Her, Kylie A. Shen, Katie Rothamel, Kasey R. Hutt, Allan C. Nojadera, Stephanie C. Bruns, Sergei A. Manakov, Brian A. Yee, Karen B. Chapman, Gene W. Yeo

# Reporting Summary

## Statistics

For all statistical analyses, confirm that the following items are present in the figure legend, table legend, main text, or Methods section.

| n/a | Confirmed | |
|---|---|---|
| ☐ | ☒ | The exact sample size (*n*) for each experimental group/condition, given as a discrete number and unit of measurement |
| ☐ | ☒ | A statement on whether measurements were taken from distinct samples or whether the same sample was measured repeatedly |
| ☐ | ☒ | The statistical test(s) used AND whether they are one- or two-sided *Only common tests should be described solely by name; describe more complex techniques in the Methods section.* |
| ☒ | ☐ | A description of all covariates tested |
| ☐ | ☒ | A description of any assumptions or corrections, such as tests of normality and adjustment for multiple comparisons |
| ☐ | ☒ | A full description of the statistical parameters including central tendency (e.g. means) or other basic estimates (e.g. regression coefficient) AND variation (e.g. standard deviation) or associated estimates of uncertainty (e.g. confidence intervals) |
| ☐ | ☒ | For null hypothesis testing, the test statistic (e.g. *F*, *t*, *r*) with confidence intervals, effect sizes, degrees of freedom and *P* value noted *Give P values as exact values whenever suitable.* |
| ☐ | ☒ | For Bayesian analysis, information on the choice of priors and Markov chain Monte Carlo settings |
| ☐ | ☒ | For hierarchical and complex designs, identification of the appropriate level for tests and full reporting of outcomes |
| ☐ | ☒ | Estimates of effect sizes (e.g. Cohen's *d*, Pearson's *r*), indicating how they were calculated |

*Our web collection on statistics for biologists contains articles on many of the points above.*

## Software and code

Policy information about availability of computer code

| Data collection | Data was processed similarly to the standard eCLIP pipeline, except for a few adjustments to ABC's multiplex design and library structure. For ABC data, unique molecular identifiers (UMI) were extracted using umitools, and adaptors were removed using cutadapt. Fastqs files were demultiplexed based on the 5' nucleotide barcode sequence using fastx toolkit (http://hannonlab.cshl.edu/fastx_toolkit/). ABC libraries were sequenced on the reverse strand. Therefore, reads were reverse complemented before alignment to repetitive regions, removal of multi-mapped reads, and alignment to the genomic sequences using STAR. The pipeline is available at https://github.com/algaebrown/oligoCLIP.git. UMI-tools version: 1.0.0 cutadapt 2.8 fastx_toolkit: 0.0.14 fastq_tools: 0.8 fastQC: 0.11.8 STAR: 2.7.6a samtools: 1.6 clipper: 2.1.2 |
|---|---|
| Data analysis | All code for figures can be found here: https://github.com/algaebrown/oligoCLIP.git |

For manuscripts utilizing custom algorithms or software that are central to the research but not yet described in published literature, software must be made available to editors and reviewers. We strongly encourage code deposition in a community repository (e.g. GitHub). See the Nature Portfolio guidelines for submitting code & software for further information.

## Data

Policy information about availability of data

 All manuscripts must include a data availability statement. This statement should provide the following information, where applicable:

- Accession codes, unique identifiers, or web links for publicly available datasets
- A description of any restrictions on data availability
- For clinical datasets or third party data, please ensure that the statement adheres to our policy

All code for analysis is accessible here https://github.com/algaebrown/oligoCLIP.git. eCLIP data used to compare against can be found here: https://www.encodeproject.org/eclip/
Encode HEPG2 RNA-seq data: https://www.encodeproject.org/experiments/ENCSR245ATJ/
Encode K562 RNA-seq data: https://www.encodeproject.org/experiments/ENCSR615EEK/

ABC data is available at GEO accession: GSE205536

## Human research participants

Policy information about studies involving human research participants and Sex and Gender in Research.

| Reporting on sex and gender | N/A |
|---|---|
| Population characteristics | N/A |
| Recruitment | N/A |
| Ethics oversight | N/A |

Note that full information on the approval of the study protocol must also be provided in the manuscript.

# Field-specific reporting

Please select the one below that is the best fit for your research. If you are not sure, read the appropriate sections before making your selection.

☒ Life sciences          ☐ Behavioural & social sciences          ☐ Ecological, evolutionary & environmental sciences

For a reference copy of the document with all sections, see nature.com/documents/nr-reporting-summary-flat.pdf

# Life sciences study design

All studies must disclose on these points even when the disclosure is negative.

| Sample size | No sample size calculations were performed. Encoded data was uploaded as duplicates and used as is. ABC data was performed in at least duplicates to match ENCODE standards. |
|---|---|
| Data exclusions | There are no data exclusions. |
| Replication | All replicates are included. eCLIP experiments were performed with two experimental replicates. ABC experiments performed with either two (singleplex) or three (multiplex) replicates. |
| Randomization | Allocation of experimental groups was not random, covariates, such as RNA expression levels, were controlled by the following: eCLIP experiment was normalized against a size matched input, singleplex ABC experiments underwent normalization against total RNA-seq/(rRNA-depleted RNA-seq when total RNA-seq is not available) whereas multiplexCLIP data normalize against both total RNA-seq and internal normalization using a chi squared test/fisher exact test between the other 9 RBPs in the multiplex. |
| Blinding | Because we were comparing the data of a new method to a previously establish method (eCLIP), blinding is not relevant to our experiment. |

# Reporting for specific materials, systems and methods

We require information from authors about some types of materials, experimental systems and methods used in many studies. Here, indicate whether each material, system or method listed is relevant to your study. If you are not sure if a list item applies to your research, read the appropriate section before selecting a response.

## Materials & experimental systems

| n/a | Involved in the study |
|---|---|
| ☐ | ☒ Antibodies |
| ☐ | ☒ Eukaryotic cell lines |
| ☒ | ☐ Palaeontology and archaeology |
| ☒ | ☐ Animals and other organisms |
| ☒ | ☐ Clinical data |
| ☒ | ☐ Dual use research of concern |

## Methods

| n/a | Involved in the study |
|---|---|
| ☒ | ☐ ChIP-seq |
| ☒ | ☐ Flow cytometry |
| ☒ | ☐ MRI-based neuroimaging |

## Antibodies

| Antibodies used | RBFOX2 Bethyl A300-864A<br>PUM2 Bethyl A300-202A<br>DDX3 Bethyl A300-474A<br>FAM120A Bethyl A300-899A<br>ACH11A Bethyl A300-524A<br>LIN28B Bethyl A300-588A<br>SF3B4 Bethyl A300-950A<br>EIF3G Bethyl A300-755A<br>PRPF8 Bethyl A300-921A<br>IGF2BP2 MBL RN008P<br>SLBP Bethyl A300-968A |
|---|---|
| Validation | Each antibody is searchable at this link: https://www.encodeproject.org/search/?type=AntibodyLot&status=released and was validated using the guidelines at this link: https://www.encodeproject.org/documents/fb70e2e7-8a2d-425b-b2a0-9c39fa296816/@@download/attachment/ENCODE_Approved_Nov_2016_RBP_Antibody_Characterization_Guidelines.pdf in addition to manufactures validation. |

## Eukaryotic cell lines

Policy information about cell lines and Sex and Gender in Research

| Cell line source(s) | K562 (Homo sapiens, adult 53 years, female) - ATCC<br>HEK293XT (Homo sapiens, embryonic, female) - Takara Bio |
|---|---|
| Authentication | Outside of the authentic commercial source, no authentication of cell lines were used. |
| Mycoplasma contamination | Cells were not tested for mycoplasma contamination. |
| Commonly misidentified lines (See ICLAC register) | No commonly misidentified lines were used in this study. |

