## [Peer Review File · Nature Methods]

Peer Review Information

Manuscript Title: Multiplexed transcriptome discovery of RNA binding protein binding sites by antibody-barcode eCLIP

Corresponding author name(s): Karen B. Chapman, Gene W. Yeo

Editorial Notes: n/a

Reviewer Comments & Decisions:

Decision Letter, initial version:
--

Dear Gene,

Your Brief Communication, "Multiplexed transcriptome discovery of RNA binding protein binding sites by antibody-barcode eCLIP", has now been seen by three reviewers. As you will see from their comments below, although the reviewers find your work of considerable potential interest, they have raised a number of concerns. We are interested in the possibility of publishing your paper in Nature Methods, but would like to consider your response to these concerns before we reach a final decision on publication.

We therefore invite you to revise your manuscript to address these concerns. We think your revisions should focus on proper controls and validations to ensure the data quality compared to eCLIP in addition to addressing the other technical concerns and questions and clarifying potential downsides of this approach.

* include a point-by-point response to the reviewers and to any editorial suggestions

* please underline/highlight any additions to the text or areas with other significant changes to facilitate review of the revised manuscript

* address the points listed described below to conform to our open science requirements

* ensure it complies with our general format requirements as set out in our guide to authors at www.nature.com/naturemethods

* resubmit all the necessary files electronically by using the link below to access your home page

[Redacted] This URL links to your confidential home page and associated information about manuscripts you may have submitted, or that you are reviewing for us. If you wish to forward this email to co-authors, please delete the link to your homepage.

We hope to receive your revised paper within three months. If you cannot send it within this time, please let us know. In this event, we will still be happy to reconsider your paper at a later date so long as nothing similar has been accepted for publication at Nature Methods or published elsewhere.

OPEN SCIENCE REQUIREMENTS

REPORTING SUMMARY AND EDITORIAL POLICY CHECKLISTS

Please note that these forms are dynamic ‘smart pdfs’ and must therefore be downloaded and completed in Adobe Reader. We will then flatten them for ease of use by the reviewers. If you would like to reference the guidance text as you complete the template, please access these flattened versions at <http://www.nature.com/authors/policies/availability.html>.

IMAGE INTEGRITY

DATA AVAILABILITY

All novel DNA and RNA sequencing data, protein sequences, genetic polymorphisms, linked genotype and phenotype data, gene expression data, macromolecular structures, and proteomics data must be deposited in a publicly accessible database, and accession codes and associated hyperlinks must be provided in the “Data Availability” section.

To further increase transparency, we encourage you to provide, in tabular form, the data underlying the graphical representations used in your figures. This is in addition to our data-deposition policy for specific types of experiments and large datasets. For readers, the source data will be made accessible

directly from the figure legend. Spreadsheets can be submitted in .xls, .xlsx or .csv formats. Only one (1) file per figure is permitted: thus if there is a multi-paneled figure the source data for each panel should be clearly labeled in the csv/Excel file; alternately the data for a figure can be included in multiple, clearly labeled sheets in an Excel file. File sizes of up to 30 MB are permitted. When submitting source data files with your manuscript please select the Source Data file type and use the Title field in the File Description tab to indicate which figure the source data pertains to.

Please include a “Data availability” subsection in the Online Methods. This section should inform readers about the availability of the data used to support the conclusions of your study, including accession codes to public repositories, references to source data that may be published alongside the paper, unique identifiers such as URLs to data repository entries, or data set DOIs, and any other statement about data availability. At a minimum, you should include the following statement: “The data that support the findings of this study are available from the corresponding author upon request”, describing which data is available upon request and mentioning any restrictions on availability. If DOIs are provided, please include these in the Reference list (authors, title, publisher (repository name), identifier, year). For more guidance on how to write this section please see: <http://www.nature.com/authors/policies/data/data-availability-statements-data-citations.pdf>

MATERIALS AVAILABILITY

****Please add a supplementary protocol****

SUPPLEMENTARY PROTOCOL

To help facilitate reproducibility and uptake of your method, we ask you to prepare a step-by-step Supplementary Protocol for the method described in this paper. We [encourage authors to share their step-by-step experimental protocols](https://www.nature.com/nature-research/editorial-policies/reporting-standards#protocols) on a protocol sharing platform of their choice and report the protocol DOI in the reference list. Nature Research's

Protocol Exchange is a free-to-use and open resource for protocols; protocols deposited in Protocol Exchange are citable and can be linked from the published article. More details can found at www.nature.com/protocolexchange/about.

ORCID

Nature Methods is committed to improving transparency in authorship. As part of our efforts in this direction, we are now requesting that all authors identified as 'corresponding author' on published papers create and link their Open Researcher and Contributor Identifier (ORCID) with their account on the Manuscript Tracking System (MTS), prior to acceptance. This applies to primary research papers only. ORCID helps the scientific community achieve unambiguous attribution of all scholarly contributions. You can create and link your ORCID from the home page of the MTS by clicking on 'Modify my Springer Nature account'. For more information please visit please visit www.springernature.com/orcid.

Sincerely,
Rita

Rita Strack, Ph.D.
Senior Editor
Nature Methods

Reviewers' Comments:

Reviewer #1:

Remarks to the Author:

This manuscript describes a new approach to multiplex CLIP analysis. Such an approach is extremely valuable to the community and will greatly enhance the comparative analysis of RNA binding proteins.

The data presented do a good job of demonstrating overall robustness of the approach, however, it would be useful for the authors to comment on a few additional points to guide researchers who might want to use this method.

First, what is the limitation in accessing/developing barcoded antibodies.

Secondly, in the data obtained in this study, do the authors observe any bias in the sequence of reads that is efficiently ligated to the barcodes? In other words, are certain terminal residues or sequence features over or under-represented in the RNA-Seq output?

Third and finally, in their comparison to eCLIP data the authors downsample the eCLIP data, presumably because the read depth for any individual RBP is less as one multiplexes samples. Can the authors say something about how much information is lost by the reduced read-depth and there is a guideline for what the cost/benefit is in how many or which RBPs are sampled at the same time (i.e. should one match expression-levels or specificity of binding? How many reads per RBP is needed to assume representative coverage of binding sites?)

Reviewer #2:

Remarks to the Author:

The authors report an update of the eCLIP protocol with proximity RNA adaptor ligation, to address issues of reproducibility with a significant reduction of hands-on time and scalability. The main asset of this protocol is the capacity to multiplex and the aggregated processing of samples. This method is a very welcome addition to the variety of *CLIP protocols with potential for high impact and use in a variety of applications. The method was performed and benchmarked on well-binding classical RNA-binding proteins RBFOX2 and SLBP and scalability tested on multiple RBPs with known target regions. Although the work is of great importance and the underlying idea is very interesting, it unfortunately lacks scientific controls and proper benchmarking.

1) Controls

The original eCLIP method underlies vast background noise and variation which was controlled by size-matched input (SMI) controls. SMI controls display complex and non-linear background behaviour (see lincRNAs like MALAT1, snoRNAs etc). This study infers through the elimination of SDS-PAGE size selection, transfer, etc. background issues are resolved and that RNA-seq can be used for normalisation which leads to target selection. Unfortunately, the authors must understand that this claim along with removal of all traditional controls together with a limited amount of replicates is worrisome and needs to be addressed. It is not given that total RNA-seq is an appropriate control, this must be discussed and supported by scientific observations in a reasonable amount of replication.

Therefore I would like to raise following questions:

Does ABC-IPs contain any enrichment of special RNAs like tRNA, lincRNAs (like MALAT1), snoRNAs?

What is the list of 100 top targets in SLBP-ABC (Figure 1G)? SLBP and RBFOX2 are extremely established CLIP RBPs. Why do stem-loops containing histone genes are not top hits for one replicate? What are the other targets? Which RNAs and RNA types were removed in the preprocessing of the analysis?

How many of eCLIP IP peaks which were removed using SMI controls are also found in ABC-CLIP?

Alternatively, a comparison of peaks detected in SMI, IP and ABC could be done. Another approach would be to remove all Histone Genes from ABC-CLIP and display a correlation (including non-coding RNAs!).

2) Why did the authors exclude any controls for antibody specificity? No crosslink or IgG controls? Why do the authors assume that barcoded antibodies do not needed to be controlled for?

3) The normalisation with RNA-seq is not appropriately documented and described. Git repository <https://github.com/algaebrown/oligoCLIP.git> does not exist.

4) The original eCLIP protocol has known chemistry issues with adapter oligorimerization which was bioinformatically dealt with a second trimming step. The proximity ligation incidentally should deal with this issue. Please confirm that those issues do not arise with ABC CLIP.

5) Replicates

Unfortunately, the *CLIP field displays bad scientific practice using a minimal amount of replicates. False positives due to biological variation can only be appropriately controlled for with biological replicates. There is hardly proof to support the authors claims to minimize sample-to-sample variation, it definitely cannot be judged using duplicates which yet alone display vast differences in library sizes. Analysing each replicate individually, specially with different library sizes, and then (eventually) overlapping is a questionable practice, especially with issues listed in 1. Practices from CHIP-seq are not necessarily appropriate for CLIP, because CLIP background is dependent on transcript level and unknown effects from crosslinking or other steps.

Also, this leads to confusion what data and binding sites were used in Figure 1D. Ideally, an analysis of RBP binding sites should lead to a list of binding sites with a measure of confidence along with a measure of effect size. Peak callers like PureCLIP can work with replicates and controls would provide you with such measures.

Due to the nature of comparing ABC-CLIP to eCLIP and to make analysis comparable, I would suggest to leave analysis with CLIPPER as is and to appropriately report methods, scripts and show all raw data of replicates (Figure 2A, D) and label them appropriately.

Furthermore provide a third replicate along a proper statistical analysis of sample-to-sample variability for ABC-CLIP.

6) Can denovo motifs be found for other benchmarked RBPs? A novel method should be benchmarked thoroughly. RBFOX2 and SLBP are exceptionally simple CLIP proteins compared to the rest of ENCODE's eCLIP data sets. Additional RBPs should be benchmarked thoroughly.

7) How do the authors explain the vast differences in library sizes? A third replicate might help estimating the variances.

8) Suppl Fig. 6. Explain Overlap with RNAseq.

Pearson Correlation is not a great measure, as displayed in the scatter plot. Maybe OLOGRAM between all libraries would be a better measure.

9) Suppl Fig 8 C&D

Also display singleplex rep1 vs rep2 and multiplex rep1 vs rep2 to get an estimate of variance within the group compared to across group.

10) In Figure 1 it is not clear if the two datasets presented for ABC and eCLIP do represent two biological replicates or the two different cell lines. It is not specified to which cell line the data in Figure 1 belongs to. Same for supplementary Figures 1 and 2.

11) Anti Body

Does the antibody barcoding effect the efficiency of the anti body? Is the barcoding site consistent?

12) In the initial ENCODE eCLIP release, to my knowledge, CLIPPER's Poisson filter settings were altered. If you compare ABC to ENCODE's releases, make sure that differences in settings are transparent.

13) Please show eCLIP results of the analysis performed in S2B in addition to ABC data.

14) Can the authors please expand on where the differences between ABC and eCLIP observed in Fig 1D come from?

15) Figure 2E. The overlap between the ABC and eCLIP profiles of FAM120A is not very consistent. Can the authors discuss potential reasons for this result?

16) Materials and methods: Could the authors please be more specific in the passage on the use of MyOne Silane beads for clean-up.

17) An in-depth description of qPCR quantification and library amplification should be provided.

18) The difference between the input versus internal normalization is very important. Could the information please be moved from supplemental note 2 to the main text? In addition, please cite the source of the RNA-seq data used for input control. Could the authors specify how important the internal normalization is? In case ABC is used in a non-multiplexed set up, is the Input normalization using RNA-seq sufficient and comparable to the use of an SMI control in eCLIP (in this case missing internal normalization)?

-) Supplemental Table 1:

Please report a table with number of reads, number of reads after adapter trimming, average sizes of libraries, number of reads discarded by RepBase alignment, and number of reads aligned, number of reads deduplicated etc.

Typo: reoved -> remove

-) "comparable read density between ABC and eCLIP at RBFOX2" Is there a statistical measurement?

-) Supplementary Figure 1: Please show the same plot for SLBP

-) Change labels of plots from "ENCODE" to "eCLIP" or "ENCODE eCLIP"

-) Label the replicates in plots

-) Supplemental Fig. 4: IDR pekas

-) Make Figure 1C and 1E comparable: Percentage or Fraction, change order

-) Figure 1F, mean relative information: explain

RBFOX2 and SLBP are great binders, better than most of ENCODE III eCLIP libraries and have rather a better

-) Check numbering of supplementary figures in Text

-) Please specify if the data in Figure 1 was derived from multiplexed or separately processed samples.

-) The authors should stay consistent in the order they present their results. Please start with either the eCLIP or the ABC data (see Fig 1 B/E vs C/D).

Reviewer #3:

Remarks to the Author:

The authors describe an innovative approach for cross-linking immunoprecipitation (CLIP) of RNA-binding proteins (RBPs) using DNA barcoded antibodies. Using this approach, several RBPs can be IPed in parallel, and the authors show evidence that it works to similar efficiency as previously established eCLIP

or related methods. Furthermore, the tedious step for separation of crosslinked conjugates on SDS-gels and recovering those complexes from nitrocellulose membranes is left-out, which makes the entire procedure more efficient and less-prone to researcher's bias. While the latter has been implemented in other related methods, this work adds further evidence that this tedious step may not be necessary.

Overall, ABC is an interesting method that will certainly attract the interest of many researchers. The manuscript is well written, concise, and comes straight to the point. Some of the key inventions though could need a more detailed description on the methodological aspects and procedures for optimization could be described. Some references to controls could also be included

Specific points:

1. The methodology section should be expanded and include all 'numbers' in regard of amounts of reagents, volumes and concentrations across all steps. For instance, how many cells were used in the assay? Concentration of extract used for IPs (ODs/protein amount).
2. Related to above, the authors should add more details regarding the protocol such as speed of rotating, if mixing is required or not during RNase I digest, etc. It may be helpful to the reader to be given a detailed step-by-step procedure in the Supplement.
3. Barcoding antibodies: This is a key feature of the protocol but information for this step is limited. One wonders about the efficiency of barcoding of antibodies? Have the oligos and proteins been titrated and how is the efficiency of barcoding evaluated? What type of controls should be included?
4. What are the limitation of ABC? Besides the obvious advantages, the authors should discuss limitations of the technique. This may including barcoding problems, specificity of the antibodies and omitting the gel-separation step.
5. Fig. 1a. The antibodies are barcoded prior to IP. However, the figure scheme suggests the reverse. This should be modified to follow the experimental procedure.

Author Rebuttal to Initial comments
--

Responses to Reviewer #1:

Remarks to the Author:

This manuscript describes a new approach to multiplex CLIP analysis. Such an approach is extremely valuable to the community and will greatly enhance the comparative analysis of RNA binding proteins.

The data presented do a good job of demonstrating overall robustness of the approach, however, it would be useful for the authors to comment on a few additional points to guide researchers who might want to use this method.

We thank the reviewer for recognizing that the approach is “extremely valuable to the community” and will “greatly enhance the comparative analysis of RNA binding proteins.”

First, what is the limitation in accessing/developing barcoded antibodies.

Barcoding of antibodies itself is simple, there are multiple working protocols. To help readers with their own barcoding, in our revised manuscript we describe clear steps for barcoding antibodies in the Methods section (copied below for the reviewer). In our experience, we have found that the major bottleneck to working antibodies for RBPs are that they have to be IP-grade in CLIP wash conditions (Sundararaman et al. 2016). Once users have an IP-grade antibody in hand, barcoding them should not be challenging.

“Oligo Barcoding Prep:

100 μ l of 100 μ M oligo barcode (IDT) in PBS and 10 μ l of 10 mM azide-NHS (Click Chemistry Tools cat# 1251-5) in DMSO were rotated at room temperature for two hours. Unreacted azide was removed by buffer exchanging into PBS using Zeba desalting columns (Thermo cat# 89883) following the manufacturer’s protocol. Azide labeled barcodes were stored at -20°C.

Antibody Barcoding:

20 μ g of antibodies were diluted to 70 μ L in PBS (enough for 4 IPs), and the buffer was exchanged into PBS using Zeba desalting columns following the manufacturer’s protocol. 10 μ L of 10 mM DBCO-NHS (Click Chemistry Tools cat# A134-10) was then added to the antibodies and allowed to rotate at room temperature for one hour (Lovci et al. 2013). Unreacted DBCO-NHS was removed by buffer exchanging into PBS using Zeba desalting columns and stored at 4°C.

6.65 μ L azide containing barcodes were then reacted with all the DBCO labeled antibodies (~70 μ L). Make sure to note which barcode is attached to each antibody. The mixture was allowed to rotate overnight at room temperature. Labeled antibodies were stored at 4°C and assumed to be 20 μ g and used as is.”

Secondly, in the data obtained in this study, do the authors observe any bias in the sequence of reads that is efficiently ligated to the barcodes? In other words, are certain terminal residues or sequence features over or under-represented in the RNA-Seq output?

To address this question, we first clarify the read structure with a *new* panel in supplementary Fig. 1B. Due to the presence of the UMI between the barcode sequences and the RNA insert, the barcode sequence itself is unlikely to produce a ligation bias. We have clarified this to readers by adding "The UMI was designed to be between the barcode sequence and RNA insert to avoid any ligation bias due to defined barcode sequences." to the figure legend.

To assure the reviewer that this was investigated, we sampled 10,000 reads from both eCLIP and ABC libraries, and selected the sequences at the 5' and 3' end. As shown in Reviewer Fig. 1 (A,B) below, compared to RBFOX2 eCLIP, RBFOX2 ABC (singleplex) shows a very minor GC-bias towards the read end, while for eCLIP the 3' end (5' end of read 2) show a slight bias in T-rich sequences but these are extremely small differences. To further attest that the barcode itself introduces no ligation biases, we compared RBFOX2 and IGF2BP2, which are prepared in the same library, but uses a different barcode. From Reviewer Fig. 1 (C,D), both libraries show the same GC preferences. Based on the above analysis and our design of the read structure, we are certain that barcodes are unlikely to produce biases in the data.

Reviewer Figure 1

Third and finally, in their comparison to eCLIP data the authors downsample the eCLIP data, presumably because the read depth for any individual RBP is less as one multiplexes samples.

Can the authors say something about how much information is lost by the reduced read-depth and there is a guideline for what the cost/benefit is in how many or which RBPs are sampled at the same time (i.e. should one match expression-levels or specificity of binding? How many reads per RBP is needed to assume representative coverage of binding sites?)

The reason we downsampled the eCLIP data was because in multiplexed ABC, the RBPs expectedly yielded different numbers of sequencing reads, that is likely due to differences in the expression and availability of the RNA substrates to crosslinking and the variation in efficiency of the different antibodies for IP. Keep in mind that we can always sequence deeper unless there's unusually poor representation by one or two RBPs in the multiplex. Then it makes sense to run those RBPs on their own (which is also feasible by ABC). To be fair, as this is the first time we (as a field) are multiplexing CLIPs, we are still learning about which RBPs are best sampled at the same time and some aspect of this is best empirically defined. In previous papers (Van Nostrand et al. 2016, 2020) we had computed marginal increase in information versus marginal increase in sequencing depth which we can compute for each RBP, however we do note that we often do not reach the depth of sequencing required for full saturation of all binding sites, but sufficient depth such that we gain molecular and functional insights which are the goals for most of the published studies. Therefore, in this study we elected to keep our sequencing depth in-line with current ENCODE standards and aimed for ~25 million reads per barcode for the total pool. To help clarify these points we have added the following to the main text as well as added Supplementary Note 2:

"As each antibody in a multiplexed reaction immunoprecipitates different amounts of protein-RNA targets due to factors such as protein expression and expression levels of RNA targets, we expectedly observe non-uniform coverage across barcodes. For a fair comparison, we computationally downsampled the uniquely mapping eCLIP reads to the same sequencing depth as the demultiplexed ABC libraries (Supplementary Table 2, Supplementary Note 2)."

"Following eCLIP guidelines, libraries were sequenced at ~25 million reads per barcode (ie 250 million reads for a 10 plex)."

Supplementary Note 2. Guidance for pooling and sequencing depth

We have observed varying amounts of reads assigned to each barcode, despite pooling an equal number of beads and antibodies. This is likely due to each RBP having varying levels of expression, expression of a given RBP's target RNAs, and the effectiveness of the IP antibody for that RBP. Future users should be mindful that antibodies need to be CLIP grade and, at this time, we suggest empirically tested and adjusted if equal coverage is required. Based on eCLIP/ENCODE guidelines we found that sequencing each sample following the formula (# of barcodes X 25 Million reads) provided sufficient depth of each RBP to recover their known biological binding preferences. However, for many RBPs this is insufficient depth to reach complete saturation of binding sites. Each RBP will have to be empirically tested to determine the correct sequencing depth to reach complete saturation."

Responses to Reviewer #2:

Remarks to the Author:

The authors report an update of the eCLIP protocol with proximity RNA adaptor ligation, to address issues of reproducibility with a significant reduction of hands-on time and scalability. The main asset of this protocol is the capacity to multiplex and the aggregated processing of samples. This method is a very welcome addition to the variety of *CLIP protocols with potential for high impact and use in a variety of applications. The method was performed and benchmarked on well-binding classical RNA-binding proteins RBFOX2 and SLBP and scalability tested on multiple RBPs with known target regions. Although the work is of great importance and the underlying idea is very interesting, it unfortunately lacks scientific controls and proper benchmarking.

We thank the reviewer for recognizing the importance and interest in the idea and we address his/her concerns on benchmarking and controls here.

1) Controls

The original eCLIP method underlies vast background noise and variation which was controlled by size-matched input (SMI) controls. SMI controls display complex and non-linear background behaviour (see lincRNAs like MALAT1, snoRNAs etc). This study infers through the elimination of SDS-PAGE size selection, transfer, etc. background issues are resolved and that RNA-seq can be used for normalisation which leads to target selection. Unfortunately, the authors must understand that this claim along with removal of all traditional controls together with a limited amount of replicates is worrisome and needs to be addressed. It is not given that total RNA-seq is an appropriate control, this must be discussed and supported by scientific observations in a reasonable amount of replication.

We agree with the reviewer that it is not a given that total RNA-seq is an appropriate control. We also agree that the original eCLIP/CLIP methodologies come with background noise and variation that the size-matched input (SMI) control we introduced (Van Nostrand 2016 Nature Methods) helped address. In fact, we want to be careful to remind ourselves and the reviewer that the utility of SMI is really in prioritizing binding sites of a given RBP that are enriched for that RBP over other RBPs (matched in the same size on the SDS-PAGE gel). The enrichment score (IP over SMI) is really "a rank-based metric for specificity of binding" (see Fig. 3 of our review (Wheeler, Van Nostrand, and Yeo 2018) explaining this at depth)). To this end, if there are sufficient numbers of eCLIP/CLIP datasets that we can use as background against specific RBPs being interrogated, interestingly, they functionally serve the same role as the SMI experiment.

Therefore I would like to raise following questions:

Does ABC-IPs contain any enrichment of special RNAs like tRNA, lincRNAs (like MALAT1), snoRNAs? What is the list of 100 top targets in SLBP-ABC (Figure 1G)? SLBP and RBFOX2 are extremely established CLIP RBPs. Why do stem-loops containing histone genes are not top hits for one replicate? What are the other targets? Which RNAs and RNA types were removed in the preprocessing of the analysis?

We do observe highly abundant small RNA species including snoRNAs (about 5-10%) in ABC-IPs (including SLBP ABC). These are also sometimes observed in CLIP/parCLIP/eCLIP IPs as these highly abundant small RNAs sometimes survive the wash, membrane and transfer steps. But the SMI control normalizes these out as experimental background and prioritizes the binding sites most specific for the RBP of interest. For ABC, when we either normalize SLBP (for example) to the complement control (CC) which is the other RBPs in the 10-plex OR SLBP eCLIP SMI, the ranking of histone genes among the top 100 ranked targets are comparable to eCLIP (with SMI normalization). We have now depicted this very clearly with a new Supplementary Fig. 8E panel. We have also included the same ENCODE thresholds for retaining enriched peaks with $p < 0.001$ and 8-fold enrichment over background (CC, SMI or RNA-seq). We also updated our previous notion in the revised manuscript to indicate that RNA-seq is less superior to CC or SMI as a way to rank-prioritize binding sites. We have also included the list of top 100 genes for SLBP ABC and eCLIP in the Supplementary Table 1.

New Supplementary Figure 8E

We add text in our revised manuscript to clarify these points, including:

“Interestingly, when all the RBPs in the multiplex set (which did not contain SLBP) were used as the CC in analysis of the SLBP singleplex experiment, peaks in histone peaks are prioritized higher in rank, compared to using RNA-seq as background (Supplementary Fig. 8D & E).”

How many of eCLIP IP peaks which were removed using SMI controls are also found in ABC-CLIP? Alternatively, a comparison of peaks detected in SMI, IP and ABC could be done. Another approach would be to remove all Histone Genes from ABC-CLIP and display a correlation (including non-coding RNAs!).

We would like to thank the reviewer for suggesting this analysis. Comparing the ABC peaks ranked by different methods to eCLIP IP peaks removed using SMI revealed that ranking peaks for RBP specificity using a simple non-parametric statistic such as a chi-square test works very well (new Supplementary Fig. 8B & C). For most ABC 10-plex libraries, RNA-seq normalized peaks contain more peaks that are removed by SMI in eCLIP. Peaks prioritized by complement control show a lower percentage of SMI-removed peaks. This indicates that RNA-seq is less effective compared to either using SMI which accounts for the background introduced by the CLIP protocol, or using CC as a rank-based metric for RBP specificity.

New Supplementary Fig. 8B & C

2) Why did the authors exclude any controls for antibody specificity? No crosslink or IgG controls? Why do the authors assume that barcoded antibodies do not needed to be controlled for?

We successfully deconvoluted our 10 RBPs from the multiplex ABC experiments (from the same tube) and showed that the binding sites compared well with published eCLIP experiments. This demonstrated that the specificity of the antibodies (post-barcoding) remained intact. However, to further demonstrate that barcoding does not affect IP efficiency, we have now included a western blot of a RBFOX2 IP with and without barcoding that indicates essentially no difference in IP efficiency (newly Supplementary Fig. 2 and also in Methods).

New Supplementary Fig. 2

ABC utilizes optimized UV, IP and wash conditions from the eCLIP protocol, and previous eCLIP studies had already demonstrated these wash conditions enable efficient library preparation when UV is used to crosslink the RBP to RNA. But to address the reviewer's concern about whether UV was necessary for barcoded antibodies, we have now performed a library prep (RBFOX2) with and without UV crosslinking and observed a 5 cycle improvement by qPCR with UV, confirming expectedly that crosslinking is necessary in these conditions.

We have updated to text to reflect these experiments with the following addition:

"After observing no change in IP efficiency after antibody barcoding (Supplementary Fig. 2) single-plex ABC experiments for RBFOX2 were performed in duplicate in HEK293T cells."

"Complexity was dependent on UV crosslinking as a no UV control displayed a 32-fold decrease in library yield by qPCR."

3) The normalisation with RNA-seq is not appropriately documented and described. Git repository <https://github.com/alqaebrown/oligoCLIP.git> does not exist.

We apologize for not ensuring the Git repository (<https://github.com/alqaebrown/oligoCLIP>) was public and complete prior to submission, we have ensured that everything is now up to date. We have since added several new methods to our Git repository that we believe will be useful for the community. In particular, the complement control is conducted in `rules/chi.py`. Normalization to another library (RNA-seq/SMI) is conducted in `rules/Snake_CLIPper.py`. Both submodules are called in `Snake_PeakMain.py` that orchestrates anything from bam to different kinds of normalization. Additionally, we added a new section to the Methods section to further detail our normalization strategies.

4) The original eCLIP protocol has known chemistry issues with adapter oligomerization which was bioinformatically dealt with a second trimming step. The proximity ligation incidentally should deal with this issue. Please confirm that those issues do not arise with ABC CLIP.

The adapter oligomerization can still be an issue for ABC due to the ssDNA ligation step. However, we designed our oligo to contain a blocking modification to prevent this. Out of an abundance of caution, we did still computationally remove adapters twice, with the second time removing adapters from ~1% of reads compared to 40% on the initial pass for the first 10-plex replicate.

5) Replicates

Unfortunately, the *CLIP field displays bad scientific practice using a minimal amount of replicates. False positives due to biological variation can only be appropriately controlled for with biological replicates. There is hardly proof to support the authors claims to minimize sample-to-sample variation, it definitely cannot be judged using duplicates which yet alone display vast differences in library sizes. Analysing each replicate individually, specially with different library sizes, and then (eventually) overlapping is a questionable practice, especially with issues listed in 1. Practices from ChIP-seq are not necessarily appropriate for CLIP, because CLIP background is dependent on transcript level and unknown effects from crosslinking or other steps.

We have removed our claim about sample-to-sample variation that was based on improved Pearson correlation values between ABC and eCLIP replicates (Supplementary Fig. 9, Compare columns ABC-other rep and eCLIP-other rep)

Also, this leads to confusion what data and binding sites were used in Figure 1D. Ideally, an analysis of RBP binding sites should lead to a list of binding sites with a measure of confidence along with a measure of effect size. Peak callers like PureCLIP can work with replicates and controls would provide you with such measures.

We apologize for the confusion with Figure 1D. To add clarity around how the binding sites were determined we have updated the methods section to include "We retain binding sites that satisfy ENCODE thresholds (p-value < 0.001 and greater than 8-fold change." As requested, the Gene Expression Omnibus(GEO) contains a table/list of all binding sites with their p-values (confidence) and fold-change (effect size).

We attempted to utilize PureCLIP but its long run time (running more than 24 hours with 4 processors) is impractical and not scalable with multiplex ABC datasets. In addition, PureCLIP currently only supports 2 replicates. We are developing new algorithms and software packages to support high numbers of replicates that can run scalably for much larger datasets, anticipating easy and widespread adoption of ABC, but that is currently not in the current scope (and time) of this Methods-based Brief Communications.

Due to the nature of comparing ABC-CLIP to eCLIP and to make analysis comparable, I would suggest to leave analysis with CLIPPER as is and to appropriately report methods, scripts and show all raw data of replicates (Figure 2A, D) and label them appropriately.

We agree with the reviewer and retain analyses with CLIPPER as is and we report methods, scripts and show all raw data of replicates. Specifically, the raw (unfiltered) data for Figure 2D can be found in Supplementary Fig. 5. To add clarity we have adjusted the row names from "ABC" to "ABC (raw)". Additionally, replicates were displayed for all figures except Figure 2A; however, we have updated the figure legends to make this more explicit "Duplicates are displayed adjacent to each other.". We elected to not show both replicates for Figure 2A because it would then represent 40 different coverage files (2 10-plexes and 20 eCLIPs) and is challenging to display and easily interpret. Since all the other figures in Figure 2, including 2D, contain replicates and the fact that they show little deviation between identified RNA features across replicates we feel keeping Figure 2A the same is best.

Furthermore provide a third replicate along a proper statistical analysis of sample-to-sample variability for ABC-CLIP.

We have removed our claim around improving sample-to-sample variability. We would like to point out that the current ENCODE eCLIP datasets are in duplicates, and we computed the variation between these duplicates (Supplementary Fig. 9, Compare columns eCLIP-other rep). Computing the variation with our ABC replicates reveals similarly high levels of correlation (Supplementary Fig. 9, Compare columns ABC-other rep). Since both technologies had a high degree of correlation, adding a 3rd replicate is unlikely to change conclusions about the reproducibility of the method.

6) Can denovo motifs be found for other benchmarked RBPs? A novel method should be benchmarked thoroughly. RBFOX2 and SLBP are exceptionally simple CLIP proteins compared to the rest of ENCODE's eCLIP data sets. Additional RBPs should be benchmarked thoroughly.

We chose RBFOX2 and SLBP not because they are simple proteins to CLIP, but because they are well validated RBPs (which have also been utilized as benchmarks in (Van Nostrand et al. 2016; Brannan et al. 2021)). We did obtain de novo motifs for PRPF8 and SF3B4 (in an updated Supplementary Fig. 7), but as the reviewer knows, not all RBPs have clear primary sequence motifs. Therefore, when selecting these other RBPs that were previously analyzed by ENCODE, we did not specifically pick proteins that had clear motifs but that their antibodies were of high IP quality and that their binding sites represented well defined regional preferences within genes that highlighted their molecular function.

7) How do the authors explain the vast differences in library sizes? A third replicate might help estimating the variances.

We appreciate the reviewer's attention to detail. Even though ABC contains fewer steps than eCLIP it is still a complex multi-day protocol and producing libraries with the same depth is challenging. There are many factors such as quantification, pooling and clustering efficiency between samples run on the same flow cell that can also affect final library depth. Finally,

because ABC's sequencing ratio is dependent on the library of each IP, even if the total library of an ABC experiment is normalized, barcodes within each library may vary. It is our opinion that overall, ABC does not exhibit any more variation between libraries than eCLIP. Evidence of this is below with a table containing examples of ENCODE approved libraries and their variation. However, our observations are that the datasets remain pertinent in extracting robust and consistent biological signals (ie. motifs, metagene distributions).

RBP	Rep1		Rep2	
	ct	unique reads	ct	unique reads
PUS1	17.379374	555685	14.588949	2107550
HNRNPA1	9.850755	7249403	10.994613	3058146
PABPC4	14.477744	3869736	17.873786	888808

8) Suppl Fig. 6. Explain Overlap with RNAseq.

Pearson Correlation is not a great measure, as displayed in the scatter plot. Maybe OLOGRAM between all libraries would be a better measure.

We thank the reviewer for suggesting OLOGRAM; however, we have looked into the software package and it appears to be a means of statistically determining the overlap between two set of genomic regions. It is our opinion that the Pearson correlation values still represent a better metric for comparing read counts inside peaks between replicates and RNA-seq as this not only shows the number of overlapping peaks but the number of reads within that peak. Perhaps what the reviewer is commenting on is that the correlation with RNA-seq is low. Given that RBPs often do not bind all RNAs in a stoichiometric fashion to total RNA levels, this low correlation is actually expected.

9) Suppl Fig 8 C&D

Also display singleplex rep1 vs rep2 and multiplex rep1 vs rep2 to get an estimate of variance within the group compared to across group.

We have added these plots to now new Supplementary Fig. 10.

10) In Figure 1 it is not clear if the two datasets presented for ABC and eCLIP do represent two biological replicates or the two different cell lines. It is not specified to which cell line the data in Figure 1 belongs to. Same for supplementary Figures 1 and 2.

We apologize for the lack of clarity. The main text did include a statement that "Single RBP replicate ABC experiments for both RBPs were performed in HEK293T and K562 cells,

respectively," for Fig. 1. However, in order to prevent further confusion we also updated the figure legend to include a similar statement.

"g) Genome browser tracks showing binding sites of RBFOX2 and SLBP for duplicate ABC and eCLIP experiments. Each panel is group normalized by RPM value. Replicate RBFOX2 data was generated in HEK293T cells and SLBP data was generated from K562 cells."

"Supplementary Fig. 3: The number of unique molecules was estimated as a function of sequencing depth. After randomly sampling uniquely mapped reads from ABC, eCLIP, and iCLIP, we plot the number of uniquely mapped reads vs the number of usable reads. a) RBFOX2 (HEK293) b) SLBP (K562)"

"Supplementary Fig. 4: a) Stacked bar plots of the total number of peaks localized to each RNA feature from HEK293 cells. b) HOMER motif analysis of the significant peaks ($P < 0.001$ and >8 -fold change). Peaks were stratified by region (CDS, 3'UTR, proximal intron, or distal intron)."

11) Anti Body

Does the antibody barcoding effect the efficiency of the anti body? Is the barcoding site consistent?

As stated earlier in our response, we have now included the IP western blot with and without barcoding (new Supplementary Fig. 2). The barcoding does not seem to affect the efficiency of the antibody in IP-ing the protein-RNA complexes. Our overall analysis of the ABC 10-plex and single-plex with ENCODE eCLIP datasets confirm similarities in essentially all metrics we have used, including motif preferences, positional preferences etc. All of which suggests the barcodes yielded equivalent data.

New Supplementary Fig. 2

The conjugation of the barcoding is on any lysine on the antibody, and we suspect this is not the same site on every antibody for a given RBP. This is also true in CITE-seq and other oligo-conjugation approaches but we do not see this as a major concern.

12) In the initial ENCODE eCLIP release, to my knowledge, CLIPPER's Poisson filter settings were altered. If you compare ABC to ENCODE's releases, make sure that differences in settings are transparent.

We appreciate the reviewers' knowledge of CLIPPER settings. To ensure this was not an issue all data for this manuscript was processed with the same version of CLIPPER. We have indicated in the methods the release version of CLIPPER 2.1.2 and by default the Poisson distribution is fit with '--supralocal'

13) Please show eCLIP results of the analysis performed in S2B in addition to ABC data.

We have updated S2B, now S4B, to contain the eCLIP motif.

14) Can the authors please expand on where the differences between ABC and eCLIP observed in Fig 1D come from?

The numbers of ABC and eCLIP sites with corresponding changes once we overlapped with the KD RNA seq datasets are small. Therefore, when calculating the odds ratio we observed a minor increase in the background, for example 0 to 1 overlap, would cause a significant change in the odds ratio value. To help clarify that these technologies are detecting comparable functional RBFOX2 sites we also included the total number of sites detected.

15) Figure 2E. The overlap between the ABC and eCLIP profiles of FAM120A is not very consistent. Can the authors discuss potential reasons for this result?

We believe this inconsistency is due to the difference in total number of peaks. FAM120A produced the second most number of peaks, >2x that of other 3' UTR RBPs, because of this the chances of these peaks overlapping with other 3' UTR binding RBPs is higher. We have improved our filtering with ENCODE guidelines, to only include peaks that have a greater than 8-fold enrichment, throughout all figures, which has minimized these differences. Additionally, to confirm this was not due to our complement control normalization we plotted the same figure with RNA-seq normalizations and observed the same trend.

16) Materials and methods: Could the authors please be more specific in the passage on the use of MyOne Silane beads for clean-up.

We have updated the methods to include a section on RT clean-up using the Silane beads.

"RT Cleanup:

MyOne Silane beads were prepared by adding 5 µl beads into a fresh tube containing 25 µl RLT buffer + 0.01% Tween 20. The tube was placed on a magnet and supernatant removed and

replaced with 93 μ l RLT buffer + 0.01% Tween 20. 90 μ l of beads were then added to the pH neutralized RT cDNA and incubated at room temperature for 10 minutes. Beads were washed with 300 μ l of 80% ethanol twice. Following the last wash the beads were allowed to air dry until the beads no longer appeared wet. The cDNA was then eluted in 2.5 μ l ssDNA ligation adapter (50 μ l 100 μ M ABC i7primer, 60 μ l DMSO, 140 μ l Bead Elution Buffer) and heated to 70 °C for 2 minutes before being placed on ice."

17) An in-depth description of qPCR quantification and library amplification should be provided.

We have updated the methods to include more details around how qPCR quantification and library amplification were performed.

"PCR Quantification:

1 μ l cDNA was diluted with 10 μ l water. 1 μ l of the diluted cDNA was mixed with 2 μ l of each qPCR primer (1.25mM in water) and 5 μ l Luna qPCR Master Mix. Samples were processed on a StepOnePlus System. Final libraries were amplified with dual index Illumina primers using Next Ultra II Q5 Master Mix. If necessary to remove adapter dimers, libraries were run and extracted from a 1-2% eGel using a Qiagen Gel Extraction kit. Library were quantified by TapeStation and sequenced on an Illumina Nextseq 2000. Following eCLIP guidelines, libraries were sequenced at ~25 million reads per barcode (ie 250 million reads for a 10 plex)."

18) The difference between the input versus internal normalization is very important. Could the information please be moved from supplementary note 2 to the main text?

We have revised the following text and Supplementary Fig. 6 to help clarify.

New Supplementary Fig. 6

We assume that for a given RBP, R_i , that exists within the universe (U) of cell type (U_t) that a multiplex set ($M_i = [R_1, R_2, \dots, R_i]$) containing a number (i) of distinct RBPs will serve as a complement (R_i') as " i " approaches all possible RBPs (∞). Hence, we define using multiplexed RBPs as an input control as "complement control" (CC). Ideally, R_i should be within M_i , however, this may not be necessary.

"To identify statistically significantly enriched peaks over background, the eCLIP protocol incorporates a SMI control representing all RBP-RNA interactions (including the interrogated RBP) in the same migratory range on the gel and membrane to capture non-specific, background RNAs. Peaks that satisfied thresholds of $P < 0.001$ and were > 8 -fold higher over SMI were deemed enriched peaks for that RBP. However, as ABC removed the gel and membrane transfer steps, we reasoned that an alternative but related measure of the specificity for a RBP on a binding site is achieved by leveraging the binding information from all (other) RBPs in the multiplexed reaction as a "complement" control (CC, see Supplementary Fig. 6). To do so, we computed the chi-square statistic from a 2x2 contingency table using the observed number of reads within the region specified by a given RBP's peak, relative to the total number of reads for that RBP, compared to the number of reads from the other nine RBPs within the same region, relative to the total number of reads for those RBPs."

In addition, please cite the source of the RNA-seq data used for input control.

We have updated the data availability section to include the citation for the RNA-seq data from ENCODE.

Data Availability

All code for analysis is accessible here <https://github.com/algaebrown/oligoCLIP.git>. Data available at GEO accession: GSE205536. ENCODE eCLIP IP, size matched inputs, and RNA-seq were downloaded from <https://www.encodeproject.org/>.

Could the authors specify how important the internal normalization is? In case ABC is used in a non-multiplexed set up, is the Input normalization using RNA-seq sufficient and comparable to the use of an SMI control in eCLIP (in this case missing internal normalization)?

As we mentioned in an earlier response, we have added a new Supplementary Fig. 8 showing that normalizing ABC experiments against other ABC experiments, now called complement control, produced the best enrichment of annotated targets. In our initial single plex experiments we do demonstrate that for SLBP, RNA-seq is able produce almost as good enrichment of histone RNAs to SMI eCLIP (Fig. 1G).

-) supplementary Table 1:

Please report a table with number of reads, number of reads after adapter trimming, average sizes of libraries, number of reads discarded by RepBase alignment, and number of reads aligned, number of reads deduplicated etc.

We have included these in Supplementary Table 2.

Typo: reoved -> remove
Fixed

-) "comparable read density between ABC and eCLIP at RBFOX2" Is there a statistical measurement?

We did not calculate the statistics for this site in particular, however, the Pearson correlation between eCLIP and ABC is high across all peaks (Supplementary Fig. 9A).

-) Supplementary Figure 1: Please show the same plot for SLBP

We have now included the same plot for SLBP. However, we would like to comment that SLBP is not an ideal RBP for measuring library complexity because it binds few targets. For example, the one of the ABC SLBP libraries was less complex than eCLIP but still identified a similar amount of histone RNAs (Supplementary Fig. 8D)

-) Change labels of plots from "ENCODE" to "eCLIP" or "ENCODE eCLIP"

Updated

-) Label the replicates in plots

Similar to our points around Fig. 2A. We believe that adding 40 (10 RBPs X 2 technologies X 2 replicates) additional lines of text to each plot will make them harder to interpret. We have however updated the figure legends to make the replicates more explicit. We do provide this plot as Reviewer Fig. 2.

Figure 2.

-) supplementary Fig. 4: IDR peaks
Fixed

-) Make Figure 1C and 1E comparable: Percentage or Fraction, change order
Updated

-) Figure 1F, mean relative information: explain
RBFOX2 and SLBP are great binders, better than most of ENCODE III eCLIP libraries and have rather a better

We have included the following section to the methods to help clarify Fig. 1F.

"Metagene analysis

Here we presented two types of metagene analysis, one with raw reads (Fig. 1F), the other ones with significant peaks (Fig. 2C & D). For the raw read metagenes (Fig. 1F) we used a software called Metadensity (<https://github.com/YeoLab/Metadensity>, *manuscript in review*) that calculates the relative information content (RIC) of immunoprecipitated reads versus the background (SMI/RNA-seq). RI content here serves as an approximation of binding distribution in the transcript. For each position i in a transcript, the fraction of reads truncated at each position is compared to the fraction of truncation in background: $RI = p_i \log(p_i/q_i)$. For Fig. 1F, we calculated relative information content for every histone transcript, then averaged it."

-) Check numbering of supplementary figures in Text
Updated

-) Please specify if the data in Figure 1 was derived from multiplexed or separately processed samples.

As mentioned before, we have updated the main text and figure legend to add clarity.

-) The authors should stay consistent in the order they present their results. Please start with either the eCLIP or the ABC data (see Fig 1 B/E vs C/D).

We thank the reviewer for this suggestion and have opted to present ABC first before eCLIP.

Responses to Reviewer #3:

Remarks to the Author:

The authors describe an innovative approach for cross-linking immunoprecipitation (CLIP) of RNA-binding proteins (RBPs) using DNA barcoded antibodies. Using this approach, several RBPs can be IPed in parallel, and the authors show evidence that it works to similar efficiency as previously established eCLIP or related methods. Furthermore, the tedious step for separation of crosslinked conjugates on SDS-gels and recovering those complexes from nitrocellulose membranes is left-out, which makes the entire procedure more efficient and less-prone to researcher's bias. While the latter has been implemented in other related methods, this work adds further evidence that this tedious step may not be necessary.

Overall, ABC is an interesting method that will certainly attract the interest of many researchers. The manuscript is well written, concise, and comes straight to the point. Some of the key inventions though could need a more detailed description on the methodological aspects and procedures for optimization could be described. Some references to controls could also be included

We thank the reviewer for pointing out that the manuscript is "an interesting method that will certainly attract the interest of many researchers" and is "well written, concise and straight to the point."

Specific points:

1. The methodology section should be expanded and include all 'numbers' in regard of amounts of reagents, volumes and concentrations across all steps. For instance, how many cells were used in the assay? Concentration of extract used for IPs (ODs/protein amount).

We apologize for the confusion. We had stated in the cross linking step that we used 10 million cells per experiment. We have now restated that we used 10 million cells in the immunoprecipitation step as well. Additionally, we have substantially updated the Methods section to include all numbers.

2. Related to above, the authors should add more details regarding the protocol such as speed of rotating, if mixing is required or not during RNase I digest, etc. It may be helpful to the reader to be given a detailed step-by-step procedure in the Supplement.

We have updated our Methods section to include more details throughout the protocol. Specifically, our new paragraph(s) are highlighted in blue.

3. Barcoding antibodies: This is a key feature of the protocol but information for this step is limited. One wonders about the efficiency of barcoding of antibodies? Have the oligos and proteins been titrated and how is the efficiency of barcoding evaluated? What type of controls should be included?

We thank the reviewer for their concern regarding the barcoded antibodies and would like to refer them to the manuscript which we followed for barcoding as they address these questions ((Wiener et al. 2020) reference 31)). We have now included the IP western blot with and without barcoding (new Supplementary Fig. 2). The barcoding does not seem to affect the efficiency of the antibody in IP-ing the protein-RNA complexes. Our overall analysis of the ABC 10-plex and single-plex with ENCODE eCLIP datasets confirm similarities in essentially all metrics we have used, including motif preferences, positional preferences etc. All of which suggests the barcodes yielded equivalent data.

We have added this sentence in the text to help clarify this

"After observing no change in IP efficiency after antibody barcoding (Supplementary Fig. 2) single-plex ABC experiments for RBFOX2 were performed in duplicate in HEK293T cells."

New Supplementary Fig. 2

4. What are the limitation of ABC? Besides the obvious advantages, the authors should discuss limitations of the technique. This may including barcoding problems, specificity of the antibodies and omitting the gel-separation step.

We have updated the text to contain the following statement about pooling RBPs and its limitations. Additionally because in the current iteration we removed the SDS-PAGE gel, ABC is currently unable to resolve different banding patterns.

“We conclude that a single ABC library (from 1 tube) generates similar overall results to 10 separate eCLIP experiments (from 20 tubes), with the caveat that the coverage across the different RBPs will be determined by antibody IP efficiency and expression levels.”

“Removal of the SDS-PAGE gel does prevent ABC from resolving higher-order protein-RNA complexes that can be resolved by gel electrophoresis.”

5. Fig. 1a. The antibodies are barcoded prior to IP. However, the figure scheme suggests the reverse. This should be modified to follow the experimental procedure.

We understand the confusion with regards to the initial scheme. However, this was partly by design as the barcoding antibodies step does not need to be repeated for each experiment. To address this concern we placed an asterisk in the figure with an updated figure legend referencing that the barcode does need to be done prior to the IP but can be done in batches. We additionally added a new Supplementary Fig. 1 showing this step as well as the barcode/read structure.

Figure 1

- A) Schematic of ABC and eCLIP workflows. Yellow blocks highlight differences between the two protocols. *30-nucleotide long oligonucleotide barcodes are conjugated to IP grade antibodies using click-chemistry prior to immunoprecipitation (Supplementary Figure 1A). The oligo then acts as the 3' adapter and undergoes proximity-based ligation to RBP-protected RNA fragments bound to the IPed RBP allowing for bioinformatic deconvolution from the sequence reads.”

References

- Brannan, Kristopher W., Isaac A. Chaim, Ryan J. Marina, Brian A. Yee, Eric R. Kofman, Daniel A. Lorenz, Pratibha Jagannatha, et al. 2021. “Robust Single-Cell Discovery of RNA Targets of RNA-Binding Proteins and Ribosomes.” *Nature Methods* 18 (5): 507–19.
- Lovci, Michael T., Dana Ghanem, Henry Marr, Justin Arnold, Sherry Gee, Marilyn Parra, Tiffany Y. Liang, et al. 2013. “Rbfox Proteins Regulate Alternative MRNA Splicing through Evolutionarily Conserved RNA Bridges.” *Nature Structural & Molecular Biology*. <https://doi.org/10.1038/nsmb.2699>.
- Sundararaman, Balaji, Lijun Zhan, Steven M. Blue, Rebecca Stanton, Keri Elkins, Sara Olson, Xintao Wei, et al. 2016. “Resources for the Comprehensive Discovery of Functional RNA Elements.” *Molecular Cell* 61 (6): 903–13.
- Van Nostrand, Eric L., Gabriel A. Pratt, Alexander A. Shishkin, Chelsea Gelboin-Burkhart, Mark Y. Fang, Balaji Sundararaman, Steven M. Blue, et al. 2016. “Robust Transcriptome-Wide Discovery of RNA-Binding Protein Binding Sites with Enhanced CLIP (ECLIP).” *Nature Methods* 13 (6): 508–14.
- Van Nostrand, Eric L., Gabriel A. Pratt, Brian A. Yee, Emily C. Wheeler, Steven M. Blue, Jasmine Mueller, Samuel S. Park, et al. 2020. “Principles of RNA Processing from Analysis of Enhanced CLIP Maps for 150 RNA Binding Proteins.” *Genome Biology* 21 (1): 90.
- Wheeler, Emily C., Eric L. Van Nostrand, and Gene W. Yeo. 2018. “Advances and Challenges in the Detection of Transcriptome-Wide Protein-RNA Interactions.” *Wiley Interdisciplinary Reviews. RNA* 9 (1). <https://doi.org/10.1002/wrna.1436>.
- Wiener, Julius, Daniel Kokotek, Simon Rosowski, Heiko Lickert, and Matthias Meier. 2020. “Preparation of Single- and Double-Oligonucleotide Antibody Conjugates and Their Application for Protein Analytics.” *Scientific Reports* 10 (1): 1457.

Decision Letter, first revision:

Dear Gene,

Thank you for your chatting with me about how you plan to respond to the reviewer concerns regarding your Brief Communication, "Multiplexed transcriptome discovery of RNA binding protein binding sites by antibody-barcode eCLIP". As I mentioned, we have decided to invite you to revise your manuscript as you have outlined, before we reach a final decision on publication.

[Redacted] This URL links to your confidential home page and associated information about manuscripts you may have submitted, or that you are reviewing for us. If you wish to forward this email to co-authors, please delete the link to your homepage.

We hope to receive your revised paper within one month. If you cannot send it within this time, please let us know. In this event, we will still be happy to reconsider your paper at a later date so long as nothing similar has been accepted for publication at Nature Methods or published elsewhere.

OPEN SCIENCE REQUIREMENTS

REPORTING SUMMARY AND EDITORIAL POLICY CHECKLISTS

Please note that these forms are dynamic ‘smart pdfs’ and must therefore be downloaded and completed in Adobe Reader. We will then flatten them for ease of use by the reviewers. If you would like to reference the guidance text as you complete the template, please access these flattened versions at <http://www.nature.com/authors/policies/availability.html>.

IMAGE INTEGRITY

DATA AVAILABILITY

Please include a “Data availability” subsection in the Online Methods. This section should inform readers about the availability of the data used to support the conclusions of your study, including accession codes to public repositories, references to source data that may be published alongside the paper, unique identifiers such as URLs to data repository entries, or data set DOIs, and any other statement about data availability. At a minimum, you should include the following statement: “The data that support the findings of this study are available from the corresponding author upon request”, describing

which data is available upon request and mentioning any restrictions on availability. If DOIs are provided, please include these in the Reference list (authors, title, publisher (repository name), identifier, year). For more guidance on how to write this section please see:
<http://www.nature.com/authors/policies/data/data-availability-statements-data-citations.pdf>

CODE AVAILABILITY

Please include a “Code Availability” subsection in the Online Methods which details how your custom code is made available. Only in rare cases (where code is not central to the main conclusions of the paper) is the statement “available upon request” allowed (and reasons should be specified).

MATERIALS AVAILABILITY

SUPPLEMENTARY PROTOCOL

To help facilitate reproducibility and uptake of your method, we ask you to prepare a step-by-step Supplementary Protocol for the method described in this paper. We [encourage authors to share their step-by-step experimental protocols](https://www.nature.com/nature-research/editorial-policies/reporting-standards#protocols) on a protocol sharing platform of their choice and report the protocol DOI in the reference list. Nature Portfolio's Protocol Exchange is a free-to-use and open resource for protocols; protocols deposited in Protocol

Exchange are citable and can be linked from the published article. More details can found at www.nature.com/protocolexchange/about.

ORCID

Nature Methods is committed to improving transparency in authorship. As part of our efforts in this direction, we are now requesting that all authors identified as 'corresponding author' on published papers create and link their Open Researcher and Contributor Identifier (ORCID) with their account on the Manuscript Tracking System (MTS), prior to acceptance. This applies to primary research papers only. ORCID helps the scientific community achieve unambiguous attribution of all scholarly contributions. You can create and link your ORCID from the home page of the MTS by clicking on 'Modify my Springer Nature account'. For more information please visit please visit www.springernature.com/orcid.

Sincerely,
Rita

Rita Strack, Ph.D.
Senior Editor
Nature Methods

Reviewers' Comments:

Reviewer #1:

Remarks to the Author:

The authors have satisfactorially addressed all my concerns

Reviewer #2:

Remarks to the Author:

Thanks to the authors for addressing the comments which overall improved the quality of the manuscript. Many request have been successfully resolved, however, unfortunately, the authors failed to address major concern:

R1.1) Controls:

I am happy that the authors agree that RNA-seq is not an appropriate control. This leaves the manuscript starting with an initial proof-of-principle without appropriate controls. The value of having RNA-seq normalised ABC-CLIP however adds to highlighting how important appropriate input controls are.

Although evidence for IP efficiency of the antibody and binding to known targets (with evident accumulation of false positives) is provided, the authors fail to address the lack of negative controls for this study. They assume that barcoded antibodies, a novelty of this proposed method in use with eCLIP type protocol, are not subjected to any nonspecific binding or any effects of crosslinking or other implications. Also, the authors assume that different barcodes do have the same effects on the background signal.

For the initial proof-of-principle, IgG control or no-crosslink controls would have been easy to include. At least, the issue could be also addressed with an experiment using IgG control with and without the same barcode used for the specific antibody and testing for the specificity by silverstaining. This would rule out any non-specific pulldown by the barcoded antibodies. In case there is non-specific pulldown, sequencing would be preferential. This is a major shortcoming and needs to be addressed, preferentially for SLBP and RBFOX2.

R1.2) Complementary Controls

The size-matched input (SMI) control still resembles an input control, whereas using a rank-based metric or chi-squared test for determining signal over input is an analysis preference of the authors rather than a property of the input control. One can test the hypothesis if sufficient numbers of eCLIP dataset could be used in a similar way to the SMI control, given a proper benchmark of the method. Discussion about what are proper complementary controls and what is a good number of proteins for CC may add to the manuscript. Discuss limitations of ABC-CLIP using CC with similar binding RBPs.

R1.3) Replicates

Regarding replication, the authors refer to their own ENCODE eCLIP standard using duplicates. It is the duty of the reviewer to critically evaluate those based on 2022 statistical standards independently: From a scientific point of view, estimating variance based on transcriptome levels with high-throughput sequencing does require a minimum of three independent biological replicates. Please be aware that eCLIP/ABC-CLIP sequencing is nowhere near as reproducible or robust as e.g. bulk RNA-seq, which can be stripped to two replicates on robust cases, depending on the biological question of the study. Removing the unsupported claim of reducing sample-to-sample variation was a necessary change,

however not providing a third replicate leaves the protocol with a minimal amount of replication. A third replicate would allow for robust statistical testing and modelling, addressing biological variance. The authors at least need to add a recommendation to the manuscript for performing at least three biological replicates when performing ABC-CLIP for a protein with unknown binding preferences.

R1.4) The authors confirm that similar chemistry issues known since the 2016 release of ENCODE eCLIP libraries still occur. As a reminder, in the initial protocol, also a blocking modification should prevent adaptor oligomerization. However, all ENCODE libraries display adaptor oligomerization and oligomerization of adapter/UMI fragments, which then are bioinformatically dealt with two independent adapter removing steps in preprocessing. The oligomerizations can result in wrong unique molecular identifier (UMI) readouts, longer fragments and other complications when incompletely removed. Currently this issue is hidden in the bioinformatic preprocessing and supplementary data for a non-expert reader. ABC-CLIP displays a great improvement compared to chemistry issues in eCLIP. This needs to be discussed and quantified (e.g. % improvement) in the manuscript.

R1.5) Thank you for making the .git repository available. Many links of the Readme are broken, or lead to hidden repositories. Please be so kind as to resolve this. A not logged-in user needs to be able to access all files. To make it a great resource for the community, please expand the description beyond input and outputs. It would be recommendable to describe what steps are performed, what assumptions, what statistics are used (e.g. better description of exact use of chi-squared test) etc.

R1.6) I am concerned with the interpretation of Supp. Fig 3b. The authors state “we would like to comment that SLBP is not an ideal RBP for measuring library complexity because it binds few targets”. Comparing eCLIP to ABC-CLIP libraries is a fair comparison and not every RBP binds as broadly as RBFOX. Although after applying controls, known targets can be detected in SLBP, the protocol does not seem as scalable as eCLIP due to unknown reasons. A speculation would be that the accumulation of unspecific background probably leads to poor performance in unique mapping reads compared to eCLIP. Since the authors did not address unspecific binding, the effect is to be speculated. One could perform the same analysis for the rest of the ABC-CLIP proteins to see if SLBP is an outlier or if this is a limitation to be discussed.

R1.7) Figure 2E, The explanation the authors provide why FAM120A overlaps with other RBPs provides an interesting thought but does not explain why FAM120A eCLIP does not overlap with ABC-CLIP targets in the same manner. An analysis of targets found in multiple RBPs might explain this effect.

Re 1) Supplementary Figure 8E is a great addition to the manuscript, highlighting the importance of CC or SMI control but also shows the big variability of those methods.

Addition of Supp Fig. 8B & C are a great addition to the manuscript, again highlighting the importance of proper input controls

Re 2) was addressed above

Re 3) was addressed above

Re 4) was addressed above

Re 5) was addressed above

- Thank you for addressing changes in Figure 1D.

- we all will be looking forward to more advanced algorithms and software packages for ABC-CLIP in future

Re 6) Point was, that RBFOX2 and SLBP are so simple CLIP proteins, that even with an not appropriate control, like RNA-seq, one does find a signal. Readers should not be tempted to assume that applying ABC-CLIP with not appropriate controls will result in the same. The authors must agree that using well established and well-validated RBPs as benchmarks are not a justification to neglect proper background or negative controls and make a proper use of controls and replicates clear in the manuscript.

Re 7) It is very unfortunate that even with the very welcome simplification of the protocol, drastic variation in library sizes can be seen. If the authors would address this in future, this would be welcomed by the field.

Re 8) ok

Re 9) Thank you for updating Supp Fig 10.

Re 10) Thank you for updating the Fig/legends

Re 11) Thank you for clarifying

Re 12) Thank you for making this transparent

Re 13) Thank you for including eCLIP data, this makes a nice comparison

Re 14) Thank you for including total sites, difference may also come from false positives (RNA-seq as a control)

Re 15) was addressed above

Re 16) Thank you for including details

Re 17) Thank you for including details

Re 18) Thank you for adding additional details

Thank you for adding reference to the RNA-seq data

Thank you for addressing all other minor comments.

Reviewer #3:

Remarks to the Author:

The authors have adequately addressed our concerns in this revised version. This is an interesting study!

Author Rebuttal, first revision:

Reviewer #2:

Remarks to the Author:

Thanks to the authors for addressing the comments which overall improved the quality of the manuscript. Many request have been successfully resolved, however, unfortunately, the authors failed to address major concern:

R1.1) Controls:

I am happy that the authors agree that RNA-seq is not an appropriate control. This leaves the manuscript starting with an initial proof-of-principle without appropriate controls. The value of having RNA-seq normalised ABC-CLIP however adds to highlighting how important appropriate input controls are.

We agree on the importance of appropriate controls. As demonstrated in Supplementary Figure 9E, we show that RNA-seq *can* serve as a control with improved prioritization of SLBP binding sites compared to not having *any* controls. However, that figure also indicates that both our CC method and SMI offer improved prioritization over RNA-seq as normalization, which is why we recommend those methods for subsequent comparisons (Figure 2). Furthermore, analyses at the raw read level (Figure 1C and 1E), positional distribution level (Supplementary Figure 9A, D, E) and binding site level (Figure 2) show that ABC and eCLIP are comparable.

Although evidence for IP efficiency of the antibody and binding to known targets (with evident accumulation of false positives) is provided, the authors fail to address the lack of negative controls for this study. They assume that barcoded antibodies, a novelty of this proposed method in use with eCLIP type protocol, are not subjected to any nonspecific binding or any effects of crosslinking or other implications. Also, the authors assume that different barcodes do have the same effects on the background signal.

In our original revised manuscript, we did perform experiments with a barcoded RBFOX2 antibody to show that IP efficiency is not affected by the barcoding (Supplemental Fig. 2). And we did inspect ligation biases (as we had pointed out in our previous rebuttal to reviewer #1) which we found essentially none (Reviewer Fig. 1). And ultimately we are pleased to see that by all standard metrics, our binding sites from ABC (with CC control) are comparable to eCLIP derived peaks (with SMIinput control). The proof is in the pudding, so to speak. As an important reminder, this study articulates a new technology that encouragingly performs well in scale and can be multiplexed, and new considerations would unlikely change this conclusion.

Reviewer Fig. 1 (A,B) nucleotide frequency count of 10,000 sampled fragments 5' and 3' end for eCLIP(A) and ABC(B) libraries. (C,D) most common 5-mer count for 10,000 sampled fragments 5' and 3' end

For the initial proof-of-principle, IgG control or no-crosslink controls would have been easy to include. At least, the issue could be also addressed with an experiment using IgG control with and without the same barcode used for the specific antibody and testing for the specificity by silverstaining. This would rule out any non-specific pulldown by the barcoded antibodies. In case there is non-specific pulldown, sequencing would be preferential. This is a major shortcoming and needs to be addressed, preferentially for SLBP and RFX2.

We thank the reviewer for this suggestion and in our original revision, we did reference no-crosslink control and observed a 32-fold increase of library yield with crosslinked (Supplemental Fig. 3C). In previous eCLIP papers from our lab and many others, the use of IgG does not actually yield useful sequenceable data and when it does (with high PCR amplification), it's not often used/useful as a control for background/rank-normalization for IP reads. Again, we would like to reiterate that across the 11 different RBPs we present in this manuscript we observe the same read, peak and regional features as in eCLIP experiments.

R1.2) Complementary Controls

The size-matched input (SMI) control still resembles an input control, whereas using a rank-based metric or chi-squared test for determining signal over input is an analysis preference of the authors rather than a property of the input control. One can test the hypothesis if sufficient numbers of eCLIP dataset could be used in a similar way to the SMI control, given a proper benchmark of the method. **Discussion about what are proper complementary controls and**

what is a good number of proteins for CC may add to the manuscript. Discuss limitations of ABC-CLIP using CC with similar binding RBPs.

We would like to indicate to the reviewer that the "analysis preference" of ABC is identical to that detailed in eCLIP:

Van Nostrand 2016

"The number of eCLIP reads overlapping CLIPper-identified peaks and the number overlapping the identical genomic region in the paired SMIInput sample were counted and used to calculate fold enrichment (normalized by total usable read counts in each data set), with enrichment P-value calculated by Yates' Chi-Square test (Perl) (or Fisher Exact Test (calculated in the R statistical computing software) where the observed or expected read number was below 5), which have minimal reportable P-values of 10^{-88} (for Chi-Square) and 2.2×10^{-16} (for Fisher Exact)."

Our introduction of "using a rank-based metric or chi-squared test" is consistent with our previous description, currently used by many labs. Furthermore our results did clearly indicate that use of CC (for the multiplexed approach) and SMIInput performed similarly across ten different RBPs as shown in Supplemental Figure 9A.

We do agree with the reviewer and have added a new supplementary figure and text in the manuscript around how the number of RBPs and their variety can impact CC "While using CC to identify peaks that were enriched for specific RBPs within the pool, we obviate the separate SMI library requirement. However, it is important to consider the number and variety of RBPs when implementing CC as these can affect the ranking of protein-RNA interaction sites (Supplemental Fig. 12)."

R1.3) Replicates

Regarding replication, the authors refer to their own ENCODE eCLIP standard using duplicates. It is the duty of the reviewer to critically evaluate those based on 2022 statistical standards independently: From a scientific point of view, estimating variance based on transcriptome levels with high-throughput sequencing does require a minimum of three independent biological replicates. Please be aware that eCLIP/ABC-CLIP sequencing is nowhere near as reproducible or robust as e.g. bulk RNA-seq, which can be stripped to two replicates on robust cases, depending on the biological question of the study. Removing the unsupported claim of reducing sample-to-sample variation was a necessary change, however not providing a third replicate leaves the protocol with a minimal amount of replication. A third replicate would allow for robust statistical testing and modelling, addressing biological variance. **The authors at least need to add a recommendation to the manuscript for performing at least three biological replicates when performing ABC-CLIP for a protein with unknown binding preferences.**

We agree with the reviewer that standards should evolve and improve over time and that more replicates will always be beneficial to some degree. We are encouraged that the data provided in this manuscript displayed no increase in variation compared to eCLIP (Supplemental Figure 10). Nevertheless, to help satisfy the reviewer we did generate a third replicate ABC (10-plex) experiment, which we show to strongly correlate with the first two replicates (Supplementary Figure 10). Additionally, we have reprocessed the RNA-seq correlation without downsampling to each sample, which did improve its correlation to ABC and eCLIP, however, is still lower than the correlations of the two technologies.

As an aside, in order to formally compare variation across three samples for both ABC and eCLIP as reviewer 2 requests and control differences in user implementations, batches of cells lines, this would really require that all experiments are performed by a single user using batched biological replicates with the same batch of reagents. This would total at least another 36 eCLIP libraries and 6 ABC libraries. Since the ABC data we now present in this manuscript (equivalent to 54 eCLIP libraries, 4 single plexes and 5 10-plexes, 3-K562 and 2-HEK293) displayed a similar level of correlations between replicates and identified the correct RNA features across 11 different targets, the amount of labor and reagent and sequencing costs necessary to repeat all of these experiments is not feasible and likely will not change our conclusion that our method introduced here in this Brief Communications is an important step forward in scaling eCLIP data generation.

R1.4) The authors confirm that similar chemistry issues known since the 2016 release of ENCODE eCLIP libraries still occur. As a reminder, in the initial protocol, also a blocking modification should prevent adaptor oligomerization. However, all ENCODE libraries display adaptor oligomerization and oligomerization of adapter/UMI fragments, which then are bioinformatically dealt with two independent adaptor removing steps in preprocessing. The oligomerizations can result in wrong unique molecular identifier (UMI) readouts, longer fragments and other complications when incompletely removed. Currently this issue is hidden in the bioinformatic preprocessing and supplementary data for a non-expert reader. ABC-CLIP displays a great improvement compared to chemistry issues in eCLIP. **This needs to be discussed and quantified (e.g. % improvement) in the manuscript.**

We are unsure why the reviewer feels we have “hidden” our bioinformatic pipeline when we state we followed ENCODE analysis, which as the reviewer even states makes this issue known, and have put our entire analysis pipeline on GitHub. However, to make this point crystal clear we added “adaptors were trimmed twice” to our methods section.

R1.5) Thank you for making the .git repository available. Many links of the Readme are broken, or lead to hidden repositories. Please be so kind as to resolve this. A not logged-in user needs to be able to access all files. To make it a great resource for the community, please **expand the description beyond input and outputs**. It would be recommendable to describe **what steps are performed, what assumptions, what statistics are used** (e.g. better description of exact use of chi-squared test) etc.

We would like to thank reviewer 2 for the suggestion. We replaced the non-working link for barcode.csv and annotator. We also updated the readme to include the steps, assumptions and statistics to make it clear as shown here [<https://github.com/YeoLab/oligoCLIP>]. Thank you again for the feedback.

R1.6) I am concerned with the interpretation of Supp. Fig 3b. The authors state “we would like to comment that SLBP is not an ideal RBP for measuring library complexity because it binds few targets”. Comparing eCLIP to ABC-CLIP libraries is a fair comparison and not every RBP binds as broadly as RBFOX. Although after applying controls, known targets can be detected in SLBP, the protocol does not seem as scalable as eCLIP due to unknown reasons. A speculation would be that the accumulation of unspecific background probably leads to poor performance in unique mapping reads compared to eCLIP. Since the authors did not address unspecific binding, the effect is to be speculated. One could perform the same analysis for the rest of the ABC-CLIP proteins to see if SLBP is an outlier or if this is a limitation to be discussed.

We have now included the library complexity comparisons between the ABC multiplex and eCLIP for the other RBPs interrogated (other than SLBP) and, again, results are similar between ABC and eCLIP (Supplemental Fig. 5). We thank the reviewer for the suggestion.

R1.7) Figure 2E, The explanation the authors provide why FAM120A overlaps with other RBPs provides an interesting thought but does not explain why FAM120A eCLIP does not overlap with ABC-CLIP targets in the same manner. An analysis of targets found in multiple RBPs might explain this effect.

Figure 2E is, as the reviewer asks for, an analysis of targets across multiple RBPs and the only RBP that has peaks overlapping FAM120A are other 3'UTR binders.

Re 1) Supplementary Figure 8E is a great addition to the manuscript, highlighting the importance of CC or SMI control but also shows the big variability of those methods.

Addition of Supp Fig. 8B & C are a great addition to the manuscript, again highlighting the importance of proper input controls

We thank the reviewer for recognizing the importance of controls.

Re 2) was addressed above

Re 3) was addressed above

Re 4) was addressed above

Re 5) was addressed above

- Thank you for addressing changes in Figure 1D.

- we all will be looking forward to more advanced algorithms and software packages for ABC-CLIP in future

We thank the reviewer for his/her continued support as we develop new packages.

Re 6) Point was, that RBFOX2 and SLBP are so simple CLIP proteins, that even with an not appropriate control, like RNA-seq, one does find a signal. Readers should not be tempted to assume that applying ABC-CLIP with not appropriate controls will result in the same. The authors must agree that using well established and well-validated RBPs as benchmarks are not a justification to neglect proper background or negative controls and make a proper use of controls and replicates clear in the manuscript.

We are grateful the reviewer appreciates the choice of SLBP and RBFOX2 as well established benchmarks. We want to also remind the reviewer the other 9 RBPs we evaluated are not all well established benchmarks but all performed similarly to eCLIP analyses.

Re 7) It is very unfortunate that even with the very welcome simplification of the protocol, drastic variation in library sizes can be seen. If the authors would address this in future, this would be welcomed by the field.

If the reviewer means differences in the number of sequenced reads for the different RBPs in the multiplex, we did acknowledge this as a caveat in our text "In multiplexing, the coverage across the different RBPs will be determined by their antibody IP efficiency and expression levels." In practice and in the future, we will have ways to address this but is currently out of scope for the manuscript.

Decision Letter, second revision:

Dear Gene,

Thank you for submitting your revised manuscript "Multiplexed transcriptome discovery of RNA binding protein binding sites by antibody-barcode eCLIP" (NMEMH-BC49516B). It has now been seen by the original referees and their comments are below. The reviewers find that the paper has improved in revision, and therefore we'll be happy in principle to publish it in Nature Methods, pending minor revisions to satisfy the referees' final requests and to comply with our editorial and formatting guidelines.

Please note, we do not expect additional experiments, but we would like to see updates to the text where relevant.

TRANSPARENT PEER REVIEW

Nature Methods offers a transparent peer review option for new original research manuscripts submitted from 17th February 2021. We encourage increased transparency in peer review by publishing the reviewer comments, author rebuttal letters and editorial decision letters if the authors agree. Such

peer review material is made available as a supplementary peer review file. Please state in the cover letter 'I wish to participate in transparent peer review' if you want to opt in, or 'I do not wish to participate in transparent peer review' if you don't. Failure to state your preference will result in delays in accepting your manuscript for publication.

Thank you again for your interest in Nature Methods Please do not hesitate to contact me if you have any questions.

Sincerely,
Rita

Rita Strack, Ph.D.
Senior Editor
Nature Methods

ORCID

Reviewer #1 (Remarks to the Author):

The authors could enhance the text to add discussion regarding some of the concern and potential pitfalls in their method raised by Reviewer #2 to prepare users for aspects of the assay on which to exercise caution.

Reviewer #2 (Remarks to the Author):

re R1.1) Controls

We are discussing the initial criticism that the first experiment in the manuscript does not have an appropriate control. The authors agree that RNA-seq is not an appropriate control, which is supported by Supp. Figure 10 showing how vastly different RNA-seq is to ABC-CLIP. However, the authors argue that it is better than not having any control. Also, authors suggest that by finding what they look for with the RNA-seq 'control', it doesn't really matter. Eventually, CC control is the one which is used and it shall be compared to previous SMI controlled eCLIP results. Also, negative controls would not be feasible to generate.

Newly added Supp. Fig. 3C shows relatively high RNA content in noCL, which would have been suitable for a control library. IgG control, even low in yield, can be sequenced and binding sites with IgG coverage can be flagged and removed for analysis, or the type of contained RNA analysed. If IgG does not yield RNA, this can be shown like in Supp. Fig. 3C.

re R1.2) Supplementary Figure 12 is a great addition to this method's paper, addressing CC in practical application. Where does the data for 100 RBPs come from? Are these permutations of pooled background RBPs? What are those background RBPs?

Comment on the author's comment: the "analysis preference" of eCLIP was applied to ABC-CLIP: In eCLIP, one does a comparison of IP samples versus SMI samples (e.g. 2 vs 1). In ABC-CLIP, one compares 2 (or more) samples vs many samples. Fisher Exact or Yates' Chi-Square does not take into account biological variation nor exploits other statistical properties. For the scope of this manuscript, the comparison with eCLIP data shall be sufficient. However, as already discussed, the authors are encouraged to improve the analysis methods.

re R1.3) It is appreciated that the authors have added another replicate as a confirmation that the protocol is robust. Please update the Supplementary Table accordingly.

re R1.4) Thank you for adding a short phrase to the manuscript for transparency. Unfortunately, this point still has not appropriately been addressed, please quantify the improvement as mentioned in R1.4.

re R1.5) Thank you for updating the github repository and improving the description.

re R1.6) Thank you for comparing library complexity of the other RBPs. Except for EIF3G and PUM2, it indeed does provide comparable results.

re R1.7) ok

Minor:

- Supp Fig. 6: change label ENCODE (raw)
- Fig 1E. Add y label tick marks

Reviewer #3 (Remarks to the Author):

We have seen a further modified version of the authors and the comments to final concerns of referee 2. Referee 2 made some very good points and I believe that the authors have addressed the concerns as much as possible.

I wish to add a note on some points:

R1.1. Controls. Implementation of IgG beads as control would be wishful, however, this has been done before by this lab and other labs. As the authors state did not reveal substantial benefit for data analysis. At the end, IgG beads cannot uncover bias through chemistry and barcoding of antibodies. I think the biggest issue are with the antibodies itself as many commercial ones turn-out to be not very specific, besides issues with their storage and implemented activity loss. I assume that unspecific binding from antibodies is by far more an issue. Essentially, the authors may want to highlight that negative controls should be implemented - especially if i) other reagents are used (one shall try to be as stringent as possible) and ii) if the technique is new to a lab. Secondly, they may want to mention that antibodies can add variation and data on their specificity should be checked.

R1.3. I agree with the referee's comments that one should perform triplicates. The authors have done so for some samples in a separate assay. I think this is fair and addressed the point sufficiently. I also think that it is good if some replicates are performed with another batch and at another day. The variation from this experiment could be indicative if another lab tries to replicate the same experiment.

The other point are as said well addressed.

Author Rebuttal, second revision:

Our ref: NMETH-BC49516B

Thank you for submitting your revised manuscript "Multiplexed transcriptome discovery of RNA binding protein binding sites by antibody-barcode eCLIP" (NMETH-BC49516B). It has now been seen by the original referees and their comments are below. The reviewers find that the paper has improved in revision, and therefore we'll be happy in principle to publish it in Nature Methods, pending minor revisions to satisfy the referees' final requests and to comply with our editorial and formatting guidelines.

Please note, we do not expect additional experiments, but we would like to see updates to the text where relevant.

Reviewer #1 (Remarks to the Author):

The authors could enhance the text to add discussion regarding some of the concern and potential pitfalls in their method raised by Reviewer #2 to prepare users for aspects of the assay on which to exercise caution.

In addition to our initial statements of caveats of pooling ratios and parameters to consider before implementing CC, we added the following statement “As with all CLIP experiments it is also critical to use high quality antibodies.” to the main text.

Reviewer #2 (Remarks to the Author):

re R1.1) Controls

We are discussing the initial criticism that the first experiment in the manuscript does not have an appropriate control. The authors agree that RNA-seq is not an appropriate control, which is supported by Supp. Figure 10 showing how vastly different RNA-seq is to ABC-CLIP. However, the authors argue that it is better than not having any control. Also, authors suggest that by finding what they look for with the RNA-seq ‘control’, it doesn't really matter. Eventually, CC control is the one which is used and it shall be compared to previous SMI controlled eCLIP results. Also, negative controls would not be feasible to generate.

No comments to address

Newly added Supp. Fig. 3C shows relatively high RNA content in noCL, which would have been suitable for a control library. IgG control, even low in yield, can be sequenced and binding sites with IgG coverage can be flagged and removed for analysis, or the type of contained RNA analysed. If IgG does not yield RNA, this can be shown like in Supp. Fig. 3C.

No comments to address

re R1.2) Supplementary Figure 12 is a great addition to this method's paper, addressing CC in practical application. Where does the data for 100 RBPs come from? Are these permutations of pooled background RBPs? What are those background RBPs?

These are random selections of ENCODE RBPs from the K562 dataset. We have updated the figure legend to address this point.

“Fraction of peaks within histone mRNAs for SLBP single-plex ABC observed based on various numbers of other ENCODE K562 RBPs used in CC analysis. A) Ranked by p-value. B) Ranked by fold enrichment. Error bar represents the standard deviation across 6 random permutations of selected eCLIP datasets as CC.”

Comment on the author’s comment: the “analysis preference” of eCLIP was applied to ABC-CLIP: In eCLIP, one does a comparison of IP samples versus SMI samples (e.g. 2 vs 1). In ABC-CLIP, one compares 2 (or more) samples vs many samples. Fisher Exact or Yates’ Chi-Square does not take into account biological variation nor exploits other statistical properties. For the scope of this manuscript, the comparison with eCLIP data shall be sufficient. However, as already discussed, the authors are encouraged to improve the analysis methods.

We appreciate the reviewer finding our analysis sufficient and will continue to improve the method and analysis in future experiments.

re R1.3) It is appreciated that the authors have added another replicate as a confirmation that the protocol is robust. Please update the Supplementary Table accordingly.

We have added all replicates to the Supplementary Table.

re R1.4) Thank you for adding a short phrase to the manuscript for transparency. Unfortunately, this point still has not appropriately been addressed, please quantify the improvement as mentioned in R1.4.

We elected to not quantify this improvement as this is user dependent. Adapter content can be easily manipulated based on what ratio of ampure beads is used and/or where the library is cut and extracted from a gel. Since we did not prepare all libraries in this manuscript (eCLIP was downloaded from ENCODE), we cannot ensure all libraries were prepared with the same level of stringency and therefore do not want to mislead readers into thinking one method produced more or less adapters.

re R1.5) Thank you for updating the github repository and improving the description.

re R1.6) Thank you for comparing library complexity of the other RBPs. Except for EIF3G and PUM2, it indeed does provide comparable results.

re R1.7) ok

Minor:

- Supp Fig. 6: change label ENCODE (raw)

We have updated the (raw) label to (IP).

- Fig 1E. Add y label tick marks

We have added additional labels to Figure 1E.

Reviewer #3 (Remarks to the Author):

We have seen a further modified version of the authors and the comments to final concerns of referee 2. Referee 2 made some very good points and I believe that the authors have addressed the concerns as much as possible.

We thank the reviewer for agreeing that we have addressed concerns to the extent possible.

I wish to add a note on some points:

R1.1. Controls. Implementation of IgG beads as control would be wishful, however, this has been done before by this lab and other labs. As the authors state did not reveal substantial benefit for data analysis. At the end, IgG beads cannot uncover bias through chemistry and barcoding of antibodies. I think the biggest issue are with the antibodies itself as many commercial ones turn-out to be not very specific, besides issues with their storage and implemented activity loss. I assume that unspecific binding from antibodies is by far more an issue. Essentially, the authors may want to highlight that negative controls should be

implemented - especially if i) other reagents are used (one shall try to be as stringent as possible) and ii) if the technique is new to a lab. Secondly, they may want to mention that antibodies can add variation and data on their specificity should be checked.

We agree with the reviewer that choice of antibodies is critical for quality data and have added the following text.

“ As with all CLIP experiments it is also critical to use high quality antibodies.”

R1.3. I agree with the referee's comments that one should perform triplicates. The authors have done so for some samples in a separate assay. I think this is fair and addressed the point sufficiently. I also think that it is good if some replicates are performed with another batch and at another day. The variation from this experiment could be indicative if another lab tries to replicate the same experiment.

We thank the reviewer for their suggestion and look forward to releasing more datasets and the adoption of ABC by other labs to evaluate variation between users and labs.

The other point are as said well addressed.

Final Decision Letter:

Dear Gene,

I am pleased to inform you that your Brief Communication, "Multiplexed transcriptome discovery of RNA binding protein binding sites by antibody-barcode eCLIP", has now been accepted for publication in Nature Methods. Your paper is tentatively scheduled for publication in our February print issue, and will be published online prior to that. The received and accepted dates will be June 8 2022 and Oct 28, 2022. This note is intended to let you know what to expect from us over the next month or so, and to let you know where to address any further questions.

Over the next few weeks, your paper will be copyedited to ensure that it conforms to Nature Methods style. Once your paper is typeset, you will receive an email with a link to choose the appropriate publishing options for your paper and our Author Services team will be in touch regarding any additional information that may be required.

Your paper will now be copyedited to ensure that it conforms to Nature Methods style. Once proofs are generated, they will be sent to you electronically and you will be asked to send a corrected version within 24 hours. It is extremely important that you let us know now whether you will be difficult to contact over the next month. If this is the case, we ask that you send us the contact information (email, phone and fax) of someone who will be able to check the proofs and deal with any last-minute problems.

If, when you receive your proof, you cannot meet the deadline, please inform us at rjsproduction@springernature.com immediately.

Once your manuscript is typeset and you have completed the appropriate grant of rights, you will receive a link to your electronic proof via email with a request to make any corrections within 48 hours. If, when you receive your proof, you cannot meet this deadline, please inform us at rjsproduction@springernature.com immediately.

Once your paper has been scheduled for online publication, the Nature press office will be in touch to confirm the details.

Content is published online weekly on Mondays and Thursdays, and the embargo is set at 16:00 London time (GMT)/11:00 am US Eastern time (EST) on the day of publication. If you need to know the exact publication date or when the news embargo will be lifted, please contact our press office after you have submitted your proof corrections. Now is the time to inform your Public Relations or Press Office about your paper, as they might be interested in promoting its publication. This will allow them time to prepare an accurate and satisfactory press release. Include your manuscript tracking number NMETH-BC49516C and the name of the journal, which they will need when they contact our office.

About one week before your paper is published online, we shall be distributing a press release to news organizations worldwide, which may include details of your work. We are happy for your institution or funding agency to prepare its own press release, but it must mention the embargo date and Nature Methods. Our Press Office will contact you closer to the time of publication, but if you or your Press Office have any inquiries in the meantime, please contact press@nature.com.

If you are active on Twitter, please e-mail me your and your coauthors' Twitter handles so that we may tag you when the paper is published.

Please note that *Nature Methods* is a Transformative Journal (TJ). Authors may publish their research with us through the traditional subscription access route or make their paper immediately open access through payment of an article-processing charge (APC). Authors will not be required to make a final decision about access to their article until it has been accepted. [Find out more about Transformative Journals](https://www.springernature.com/gp/open-research/transformative-journals)

To assist our authors in disseminating their research to the broader community, our SharedIt initiative provides you with a unique shareable link that will allow anyone (with or without a subscription) to read the published article. Recipients of the link with a subscription will also be able to download and print the PDF. As soon as your article is published, you will receive an automated email with your shareable link.

Please note that you and your coauthors may order reprints and single copies of the issue containing your article through Springer Nature Limited's reprint website, which is located at <http://www.nature.com/reprints/author-reprints.html>. If there are any questions about reprints please send an email to author-reprints@nature.com and someone will assist you.

Best regards,

Rita

Rita Strack, Ph.D.
Senior Editor
Nature Methods